# Differentially Private Federated Low Rank Adaptation Beyond Fixed-Matrix

**Ming Wen** *
Fudan University
Shanghai Innovation Institute
mwen23@m.fudan.edu.cn

**Jiaqi Zhu** *
Fudan University
jiaqizhu25@m.fudan.edu.cn

**Yuedong Xu** †
Fudan University
Shenzhen Loop Area Institue
ydxu@fudan.edu.cn

**Yipeng Zhou**
Macquarie University
yipeng.zhou@mq.edu.au

**Dingding Han**
Fudan University
ddhan@fudan.edu.cn

## Abstract

Large language models (LLMs) typically require fine-tuning for domain-specific tasks, and LoRA offers a computationally efficient approach by training low-rank adapters. LoRA is also communication-efficient for federated LLMs when multiple users collaboratively fine-tune a global LLM model without sharing their proprietary raw data. However, even the transmission of local adapters between a server and clients risks serious privacy leakage. Applying differential privacy (DP) to federated LoRA encounters a dilemma: adding noise to both adapters amplifies synthetic noise on the model, while fixing one adapter impairs the learnability of fine-tuning. In this paper, we propose FedASK (Differentially Private **Fed**erated Low Rank **A**daptation with Double **SK**etching) , a novel federated LoRA framework to enable effective updating of both low-rank adapters with robust differential privacy. Inspired by randomized SVD, our key idea is a two-stage sketching pipeline. This pipeline first aggregates carefully sketched, privacy-preserving local updates, and then reconstructs the global matrices on the server to facilitate effective updating of both adapters. We theoretically prove FedASK's differential privacy guarantee and its exact aggregation property. Comprehensive experiments demonstrate that FedASK consistently outperforms baseline methods across a variety of privacy settings and data distributions. Codes are available at https://github.com/FLEECERmw/PrivacyFedLLM.

## 1 Introduction

Federated fine-tuning of Large Language Models (LLMs) presents a compelling paradigm for specializing these models on domain-specific, distributed datasets without centralizing sensitive information [41, 25, 46]. However, the sheer scale of LLMs, often involving hundreds of billions of parameters, renders full-parameter fine-tuning prohibitive for local clients in FL due to memory, computation, and communication constraints [46]. To overcome these limitations, Parameter-Efficient Fine-Tuning (PEFT) methods, particularly Low-Rank Adaptation (LoRA) [20], have gained prominence [16]. LoRA facilitates efficient adaptation by freezing the pre-trained model weights $W_0$ and training only a small set of low-rank matrices, $A \in \mathbb{R}^{r \times n}$ and $B \in \mathbb{R}^{m \times r}$ (where $r \ll \min(m, n)$).

---

*Equal contribution.
†Correspondence to Yuedong Xu.

39th Conference on Neural Information Processing Systems (NeurIPS 2025).

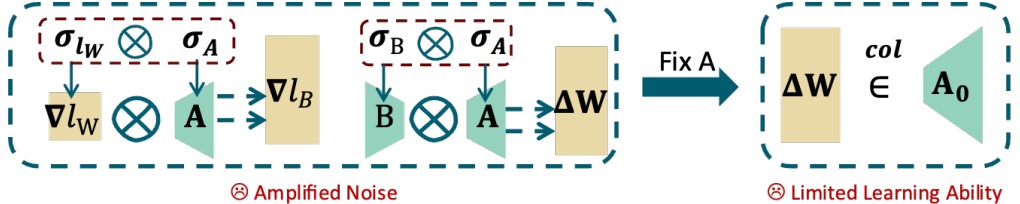

Figure 1: The *dilemma* of differential privacy with Federated LoRA: standard federated LoRA amplifies model noise, while fixing one adapter causes insufficient learnability.

The update $\Delta W = BA$ significantly reduces the number of trainable parameters, thereby alleviating local processing and communication overhead in federated LLM fine-tuning scenarios.

Ensuring privacy in federated learning is critical, especially when fine-tuning LLMs [29]. The inherent scale and complexity of LLMs lead to greater probability of encoding and revealing sensitive information from training data compared to smaller models. While raw data are not shared in FL settings, the model updates and gradients transmitted can still potentially leak private client information. Therefore, providing robust privacy protection with theoretical guarantees is essential. Differential Privacy (DP) [8, 1] serves as a principled framework to achieve this, typically by introducing calibrated noise during the training process.

However, applying DP to federated LoRA presents a fundamental trade-off. On the one hand, achieving robust privacy often requires adding noise to the gradients of low-rank matrices A and B. Despite this, the intertwined nature of these gradients leads to significant noise amplification when both matrices are perturbed. This significantly increases the magnitude of noise in the resulting $\Delta W$ update, degrading the quality of the model update [34, 22]. On the other hand, existing methods commonly address this noise amplification by fixing one of the matrices (typically A) during training. Such strategies help mitigate the noise, particularly by avoiding the detrimental quadratic noise term. However, it fundamentally restricts the representational capacity of the LoRA parameters to a predetermined subspace, thereby impairing the model's ability to adapt effectively [13, 45]. Motivated by this critical dilemma, this paper addresses the following research question: **Can we design a federated LoRA that achieves differential privacy guarantee, learnability, and communication efficiency simultaneously?**

To address this dilemma, we propose **FedASK** (Differentially Private **Fed**erated Low Rank **A**daptation with Double **SK**etching). FedASK is the first federated LoRA framework designed to enable effective updates of both low-rank matrices under robust DP, overcoming the critical noise amplification and limited learnability trade-off inherent in prior methods. Our core insight is a novel two-stage projection pipeline, inspired by randomized SVD [15].

Table 1: Federated LoRA Comparison. DP (✓/✗): Supports DP. Agg. Type: Aggregation Precision. Client Init.: Parameter state before local training (Sync=Use Global Para, Keep B=Use Local B, Fixed A=Use Initial A, Rand A=Gaussian). "$d_t$" is the input feature dimension, and "$r$" is the LoRA rank.

| METHOD | DP | AGG. TYPE | AGG. MEMORY | COMMUNICATION | CLIENT INIT. |
|---|---|---|---|---|---|
| FEDAVG | ✗ | IMPRECISE | $O(d_l r)$ | $O(K d_l r)$ | SYNC A, B |
| FLORA | ✗ | PRECISE | $O(d_l^2 + K d_l r)$ | $O(K^2 d_l r)$ | $B = \mathbf{0}, A_k^0 = \text{RAND}$ |
| FEDSA | ✗ | IMPRECISE | $O(d_l r)$ | $O(K d_l r)$ | SYNC A, KEEP LOCAL B |
| FED-FFA | ✓ | PRECISE | $O(d_l r)$ | $O(K d_l r)$ | FIX A, SYNC B |
| **FEDASK** | ✓ | PRECISE | $O(d_l r)$ | $O(K d_l r)$ | SYNC A, B |

This pipeline achieves precise and resource-efficient aggregation of local LoRA update products, avoiding the issues of directly aggregating noisy matrices. Differential privacy is guaranteed by applying DP-SGD [1] with noise addition and clipping to local updates. Critically, the global SVD-based aggregation step is not merely an aggregation, but powerfully leverages this privatized information to influence and update both global matrices A and B, facilitating comprehensive adaptation capabilities under stringent DP constraints. Our key contributions are summarized as follows.

- We introduce FedASK, the first privacy preserving LoRA framework updates both low-rank matrices and achieves robust noise suppression, privacy guarantees, and resource-efficient operations.
- We theoretically prove the DP guarantee and the precise aggregation property of FedASK.
- FedASK empirically achieves consistent performance gains over state-of-the-art baselines in federated LLM fine-tuning, delivering up to an 11.5% performance improvement on MMLU (7B model) and a 46% improvement on GSM8K (13B model) under strong differential privacy.

## 2 Preliminary

This section describes the background on LoRA fine-tuning of LLMs within the context of Federated Learning. We also introduce the concept of DP and its relevance to protecting client data.

Applying DP-SGD directly to the standard LoRA update mechanism presents challenges. Specifically, when independent DP noise is added to the gradients of both A and B, these noise [12] components interact quadratically during the construction of the LoRA update $\Delta W$. This interaction leads to significant noise amplification, as detailed in Lemma . Existing strategies commonly mitigate this noise amplification by freezing one matrix and optimizing the other . This approach successfully avoids the quadratic noise term. However, since the parameter update is constrained to a specific subspace, it limits the model's learning capability and hinders effective adaptation to the target task. Tableunderscores FedASK's distinct advantages over existing federated LoRA methods. Overcoming the adaptability limitations of fixed-matrix DP LoRA methods, FedASK enables dynamic and synchronized update of both LoRA matrices A and B under strong DP guarantees, achieving precise aggregation with resource efficiency comparable to or better than existing baselines.

### 2.1 Federated Learning with LoRA

The goal of Federated Learning with LoRA [40, 23, 43] is to minimize the global objective function:

$$F(\mathrm{W}) = \frac{1}{K} \sum_{k=1}^{K} f_k(\mathrm{W}_k, \mathcal{D}_k), \tag{1}$$

where $f_k(\mathrm{W}_k, \mathcal{D}_k) = \frac{1}{|\mathcal{D}_k|} \sum_{\xi \in \mathcal{D}_k} l(\mathrm{W}_k, \xi)$ represents the local objective function at client $k$. Here, $\mathrm{W}_k$ denotes the weight matrix of the local model, and $\mathcal{D}_k$ is the local dataset at the client $k$. The function $l(\mathrm{W}_k, \xi)$ computes the loss for a single data point $\xi \in \mathcal{D}_k$, and $f_k$ averages this loss across the entire dataset. In LoRA, instead of directly training the full weight matrix $\mathrm{W}_k$, we update it by adding a low-rank decomposition product to the pre-trained weights $\mathrm{W}_0$, and using gradient-based oracles to optimize the adapters.

$$\frac{\partial l}{\partial \mathrm{A}_k} = \frac{\alpha}{r} \mathrm{B}_k^T \frac{\partial l}{\partial \mathrm{W}_k}, \quad \frac{\partial l}{\partial \mathrm{B}_k} = \frac{\alpha}{r} \frac{\partial l}{\partial \mathrm{W}_k} \mathrm{A}_k^T, \tag{2}$$

where $\mathrm{A}_k \in \mathbb{R}^{r \times n}$ and $\mathrm{B}_k \in \mathbb{R}^{m \times r}$ are low-rank matrices with rank $r$ (typically $r \ll \min(m, n)$), and $\alpha$ is a scaling factor. These matrices $\mathrm{A}_k$ and $\mathrm{B}_k$ are the parameters learned during fine-tuning.

A straightforward approach for Federated LoRA involves clients locally training their LoRA matrices ($\mathrm{A}_k$ and $\mathrm{B}_k$) and then average them on the server [14, 28]. However, this direct averaging approach causes a misalignment with the precise global model update [38].

$$\bar{\mathrm{W}} = \frac{1}{K} \sum_{k=1}^{K} \mathrm{W}_k = \frac{1}{K} \sum_{k=1}^{K} (\mathrm{W}_0 + \frac{\alpha}{r} \mathrm{B}_k \mathrm{A}_k) \neq \mathrm{W}_0 + \frac{\alpha}{K^2 r} \sum_{k=1}^{K} \mathrm{B}_k \sum_{k=1}^{K} \mathrm{A}_k. \tag{3}$$

This discrepancy introduces undesirable cross-term noise, particularly in the presence of data heterogeneity across clients. Prior work [3, 34, 38] has recognized the same issue. Existing methods propose different strategies to address this misalignment. SLoRA [3] employs a multi-stage approach with sparse fine-tuning and SVD-based initialization, which incurs additional computational overhead. FFA-LoRA [34] adopts a simplified strategy by freezing matrix A and optimizing only B,limiting the model's expressive capability. FLoRA [38] relies on stacking and processing matrices on the server side, resulting in increased server computation and communication costs.

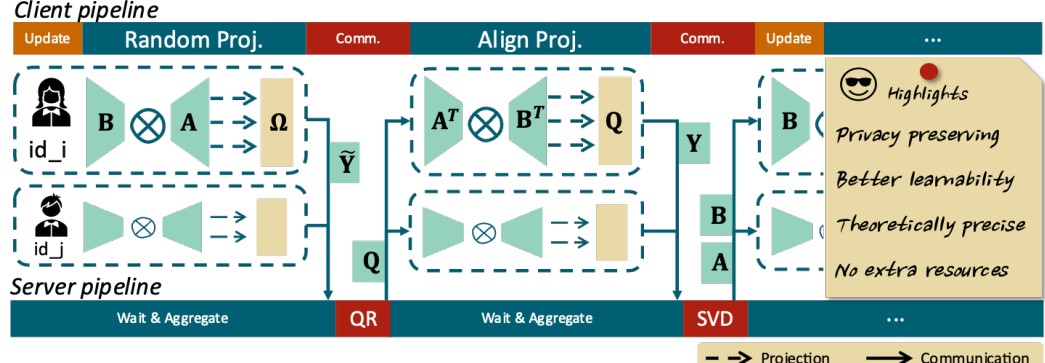

Figure 2: FedASK Pipeline.

## 2.2 DP in Federated LoRA

We hereby define differential privacy as the following [9, 11]:

**Definition 1.** *A randomized algorithm $\mathcal{M} : \mathcal{D} \to \mathcal{S}$ satisfies $(\epsilon, \delta)$-differential privacy if, for any two adjacent datasets $D, D' \in \mathcal{D}$ differing in one data point, and any output subset $S \subseteq \mathcal{S}$*

$$Pr[\mathcal{M}(D) \in S] \leq e^{\epsilon} Pr[\mathcal{M}(D') \in S] + \delta,$$

*where $\epsilon \geq 0$ controls the privacy loss and $0 < \delta < 1$ is the failure probability.*

DP theoretically ensures that this algorithm's output is nearly unaffected by the presence or absence of any single individual's data in the dataset. In the context of federated learning, DP is commonly achieved through DP-Stochastic Gradient Descent (DP-SGD)[1, 24]. This method adds calibrated noise to the gradients computed at each client before aggregation.

Applying DP-SGD directly to the standard LoRA update mechanism presents challenges. Specifically, when independent DP noise is added to the gradients of both A and B, these noise components interact quadratically during the construction of the LoRA update $\Delta$W. This interaction leads to significant noise amplification, as detailed in Lemma 1. Existing strategies commonly mitigate this noise amplification by freezing one matrix and optimizing the other [34, 22]. This approach successfully avoids the quadratic noise term. However, since the parameter update is constrained to a specific subspace, it limits the model's learning capability and hinders effective adaptation to the target task. Table 1 underscores FedASK's distinct advantages over existing federated LoRA methods. Overcoming the adaptability limitations of fixed-matrix DP LoRA methods, FedASK enables dynamic and synchronized update of both LoRA matrices A and B under strong DP guarantees, achieving precise aggregation with resource efficiency comparable to or better than existing baselines.

## 3 FedASK Framework

In this section, we presents FedASK (Differentially Private **Fed**erated Low Rank **A**daptation with Double **SK**etching) for federated fine-tuning of large language models using LoRA. We first introduce the overall FedASK framework, which contains the two-stage projection pipeline designed for efficient and precise global updates. Subsequently, we describe the integrated DP framework, which facilitates privacy-preserving training.

### 3.1 Pipeline: Two-Stage Sketching

The core innovation of FedASK lies in its efficient two-stage pipeline, which employs sketching and projection techniques to accurately compute the aggregated LoRA update $(\sum_k B_k A_k)$ while minimizing resource overhead. FedASK is motivated by the observation that weight updates in large neural networks [19, 2], particularly with methods like LoRA, exhibit a low-rank structure. This property allows their essential information to be captured efficiently in a low-dimensional subspace.

Directly computing and aggregating the full product matrices $B_k A_k$ from each client would be computationally expensive and communication-heavy, as these matrices are of the same dimension

**Algorithm 1 FedASK**

---

**Input:** Initial global weight matrix $\mathbf{W}_0$; LoRA rank $r$; LoRA scaling factor $\alpha$; Over-sketching parameter $p$; Client learning rate $\gamma$; Set of all clients $\mathcal{K}$; Communication rounds $T$; Local update steps $m$; Boolean DP flag use_DP; Input feature dimension $d_l$.

**Output:** Final global LoRA matrices $\mathbf{A}^T, \mathbf{B}^T$.

**Initialization:**

Initialize $\mathbf{A}^0 \in \mathbb{R}^{r \times d_l}, \mathbf{B}^0 \in \mathbb{R}^{d_l \times r}$; Generate $\mathbf{\Omega} \in \mathbb{R}^{d_l \times (r+p)}$.

**for** $t = 1$ **to** $T$ **do**

    Sample $\mathcal{K}_t \subseteq \mathcal{K}$; Broadcast $\mathbf{A}^{t-1}, \mathbf{B}^{t-1}, \mathbf{\Omega}$ to clients in $\mathcal{K}_t$.

    **for** each client $k \in \mathcal{K}_t$ **in parallel do**

        **if** use_DP is true **then**

            $\mathbf{A}_k^t \leftarrow \mathbf{A}^{t-1}$; $\mathbf{B}_k^t \leftarrow$ LocalUpdate_DP$(\mathbf{B}^{t-1}, \mathbf{A}_k^t, \mathcal{D}_k, m, \gamma, \alpha)$

        **else**

            $\mathbf{A}_k^t, \mathbf{B}_k^t \leftarrow$ LocalUpdate$(\mathbf{A}^{t-1}, \mathbf{B}^{t-1}, \mathcal{D}_k, m, \gamma, \alpha)$

        **end if**

        Compute $\mathbf{Y}_k^{proj} = \mathbf{B}_k^t(\mathbf{A}_k^t\mathbf{\Omega})$; Send to server.

    **end for**

    Aggregate $\mathbf{Y}_{agg}^t = \sum_{k \in \mathcal{K}_t} \mathbf{Y}_k^{proj}$; Perform QR: $\mathbf{Q}^t, \_ = $ QR$(\mathbf{Y}_{agg}^t)$.

    Broadcast $\mathbf{Q}^t$ to clients in $\mathcal{K}_t$.

    **for** each client $k \in \mathcal{K}_t$ **in parallel do**

        Compute $\tilde{\mathbf{Y}}_k^{proj} = (\mathbf{A}_k^t)^\top ((\mathbf{B}_k^t)^\top \mathbf{Q}^t)$; Send to server.

    **end for**

    Aggregate $\tilde{\mathbf{Y}}_{agg}^t = \sum_{k \in \mathcal{K}_t} \tilde{\mathbf{Y}}_k^{proj}$.

    Compute SVD of $(\tilde{\mathbf{Y}}_{agg}^t)^\top$: $\mathbf{U}, \mathbf{\Sigma}, \mathbf{V}^\top = $ SVD$((\tilde{\mathbf{Y}}_{agg}^t)^\top)$.

    Select leading $r$ components $\mathbf{U}_r, \mathbf{\Sigma}_r, \mathbf{V}_r^\top$.

    Update global LoRA matrices: $\mathbf{B}^t \leftarrow \mathbf{Q}^t\mathbf{U}_r\mathbf{\Sigma}_r^{\frac{1}{2}}$ ; $\mathbf{A}^t \leftarrow \mathbf{\Sigma}_r^{\frac{1}{2}}\mathbf{V}_r^\top$

**end for**

---

as the original weight matrices adapted by LoRA. FedASK addresses the challenge by adopting sketching principles inspired by randomized SVD [15]. Clients transmit compressed representations instead of full matrices. Since the aggregated LoRA update also maintains a low-rank structure, these sketching techniques can effectively capture its essential information, enabling the server to perform a precise reconstruction. The overall process unfolds in two stages.

**First stage: Randomized Subspace Sketching.** Within the FedASK framework, local clients perform the standard LoRA training procedures to obtain their updated local matrices, denoted as $\mathrm{B}_k^t$ and $\mathrm{A}_k^t$. To enable global aggregation, FedASK utilizes a shared random projection matrix $\Omega \in \mathbb{R}^{n \times (r+p)}$ to sketch these updated matrices. Specifically, each client computes the projection by:

$$\mathrm{Y}_k^{proj} = \mathrm{B}_k^t(\mathrm{A}_k^t\Omega). \tag{4}$$

Subsequently, this computed projection is transmitted to the server. The server aggregates these received projections and performs a QR decomposition to derive an orthonormal basis Q, which is then redistributed to the participating clients. This basis serves to capture the global singular subspace relevant to the exact aggregated $\mathrm{B}^t\mathrm{A}^t$, helping to revise the random projection of the first stage and align the local parameter to the Global Space.

**Second stage: Global Alignment Projection.** After receiving the orthonormal basis Q, the clients project their updated matrices onto this basis, yielding an updated projection:

$$\tilde{\mathrm{Y}}_k^{proj} = (\mathrm{A}_k^t)^\top ((\mathrm{B}_k^t)^\top \mathrm{Q}). \tag{5}$$

The server then aggregates these updated projections and proceeds to perform a Singular Value Decomposition (SVD) on the aggregated matrix to decompose it into its singular components. Finally, the global parameters $\mathrm{A}^t$ and $\mathrm{B}^t$ are updated based on the results of the SVD as follows:

$$\mathrm{B}^t = \mathrm{QU}\Sigma^{\frac{1}{2}}, \quad \mathrm{A}^t = \Sigma^{\frac{1}{2}}\mathrm{V}^\top. \tag{6}$$

This two-stage process enables FedASK to achieve a precise global update that is mathematically equivalent to the average of local updates $\frac{1}{K}\sum_k \mathrm{B}_k\mathrm{A}_k$ within a subspace rank $r + p$. The formal theoretical guarantee for this exact aggregation property is detailed within Section 4.

## 3.2 Differentially Private Local Updates in FedASK

To ensure the privacy of individual clients' sensitive data during the training process, we integrate Differential Privacy (DP) into our FedASK framework. As detailed in Algorithm 1, DP is applied conditionally based on the `use_DP` flag (line 6). When `use_DP` is set to true, each selected client $k \in \mathcal{K}_t$ executes the `LocalUpdate_DP` function (line 7). In the DP-enabled mode, the local update conducts on the $B_k$ matrix, while $A_k$ remains fixed for that local training phase ($A_k^t \leftarrow A^{t-1}$).

In the `LocalUpdate_DP` function, the LoRA matrix $B_k$ is updated at each step $\tau$ using the DP-SGD mechanism. The process, with gradient clipping and calibrated noise addition, is formulated as:

$$B_k^{\tau+1} = B_k^\tau - \frac{\gamma\alpha}{r}\left(\frac{\partial l}{\partial W_k^\tau} \bigg/ \max\left(1, \frac{\|\frac{\partial l}{\partial W_k^\tau}\|_2}{C}\right) + \mathcal{N}(0, \sigma^2 C^2 \mathbf{I})\right)(A^{t-1})^T, \tag{7}$$

where $\gamma$ and $\alpha$ denote the learning rate and the LoRA scaling factor, respectively.

Although $A_k$ is not directly perturbed by local noise in DP rounds, the information learned and privatized via updates to $B_k$ is not confined to global $B^t$. When the server aggregates the client projections and performs the SVD reconstruction , the resulting global factors $A^t$ and $B^t$ are both updated. The SVD step effectively decomposes the aggregated privatized information captured primarily within the $B_k$ updates and redistributes it across *both* newly formed global matrices $A^t = \Sigma^{\frac{1}{2}} V^\top$ and $B^t = QU\Sigma^{\frac{1}{2}}$.

Therefore, FedASK's unique two-stage projection and SVD-based aggregation allow the knowledge gained under differential privacy to influence the global representation captured by $A^t$ as well. The formal privacy analysis proving the $(\epsilon, \delta)$-guarantee for FedASK is presented in the following section.

# 4 Theoretical Analysis

This section highlights the theoretical analysis of FedASK, focusing on two key guarantees: robust differential privacy and precise aggregation.

## 4.1 Robust Differential Privacy Integration

Applying differential privacy to Low-Rank Adaptation (LoRA), particularly when simultaneously updating both low-rank matrices $A_t$ and $B_t$ through DP-SGD, presents a significant challenge. Lemma 1 provides a formal quantitative analysis of this noise amplification.

**Lemma 1** (Approximate Noise Power in LoRA Update with DP-SGD). *Consider LoRA parameters $A_t \in \mathbb{R}^{r \times d_l}$ and $B_t \in \mathbb{R}^{d_l \times r}$ (where $d_l \gg r$). Under DP-SGD with learning rate $\eta$, noise multiplier $\sigma$, clipping $C$, and batch size $B_{size}$, independent Gaussian noises $\xi_A, \xi_B$ (effective per-component variance $\sigma^2 C^2/B_{size}^2$) are added to gradients $\nabla A_t, \nabla B_t$, respectively. Let $\Delta W_{noise}$ be the noise component of the resulting LoRA update.*

$$\mathbb{E}[\|\Delta W_{noise}\|_F^2] \approx \underbrace{\eta^2 \frac{\sigma^2 C^2}{B_{size}^2} d_l r(\|A_t\|_F^2 + \|B_t\|_F^2)}_{\text{Term 1: Linear Noise}} + \underbrace{\eta^4 \frac{\sigma^4 C^4}{B_{size}^4} d_l^2 r}_{\text{Term 2: Quadratic Noise}} + O(\eta^4\sigma^2 + \eta^3\sigma^2). \tag{8}$$

*While Term 1 reflects standard linear noise as in DP-SGD, Term 2 arises from noise interaction and can dominate with large $\sigma$ or $\eta$, scaling with $d_l^2$. This causes the LoRA update's Signal-to-Noise Ratio (SNR) to degrade significantly faster ($1/\sigma^4$) than individual gradient SNRs ($1/\sigma^2$), revealing a critical noise amplification problem in standard DP-LoRA.*

Lemma 1 highlights a critical challenge in standard DP-LoRA: simultaneously perturbing both low-rank matrices $A_t$ and $B_t$ leads to a dominant quadratic noise term, severely degrading the LoRA update's signal-to-noise ratio and impacting model utility. FedASK involves focusing local DP-SGD updates primarily on the $B_k$ matrix, while $A_k^t$ is primarily updated on the server through the two-stage aggregation pipeline with a global SVD-based update. This strategy preempts local generation of the problematic quadratic noise term. The formal $(\epsilon, \delta)$-differential privacy guarantee for FedASK is established in Theorem 1.

**Theorem 1** (Privacy Guarantee of FedASK). *Suppose the gradient sensitivity is $C = 1$. FedASK (Algorithm 1) guarantees that the final global LoRA matrices $(A^T, B^T)$ are $(\epsilon, \delta)$-differentially private with respect to the joint dataset $D = \bigcup_{i=1}^{K} \mathcal{D}_k$ of $K$ total clients, provided the variance $\sigma^2$ of the Gaussian noise added to local gradients satisfies:*

$$\sigma^2 = \mathcal{O}\left(\frac{q_{\mathcal{D}}^2 \cdot m \cdot q_{\mathcal{K}} \cdot T \cdot \ln(2/\delta) \cdot \ln(2Tq_{\mathcal{K}}/\delta)}{\epsilon^2 \cdot K}\right),$$

*where $q_{\mathcal{K}}$ is the client sampling ratio per communication round (total $T$ rounds), and $q_{\mathcal{D}}$ is the data sampling ratio per local update (total $m$ local updates per client per round).*

Theorem 1 establishes the end-to-end $(\epsilon, \delta)$-differential privacy for FedASK. Noise variance $\sigma^2$ scales consistent with established DP-SGD analyzes in federated learning [32]. The derivation rigorously tracks privacy loss using Rényi Differential Privacy (RDP), briefly with the following arguments. (1)Prove the RDP guarantee for two sketches $Y_k^{\text{proj}}$ and $\tilde{Y}_k^{\text{proj}}$ released by a client $k$ within a single communication round, relative to its local data $\mathcal{D}_k$; (2) convert this per round, per client RDP into an intermediate $(\epsilon_0, \delta_0)$-DP guarantee; (3) applying advanced composition theorems for $(\epsilon, \delta)$-DP over $T$ communication rounds; and (4) incorporate privacy amplification from client subsampling (ratio $q_{\mathcal{K}}$) to achieve the final stated $(\epsilon, \delta)$-DP guarantee. The detailed proof is provided in Appendix A.2.

### 4.2 Aggregate Precision Guarantee

Aggregating local updates in federated LoRA presents a key challenge: achieving high accuracy while minimizing communication and computational overhead. Unlike conventional methods introducing approximation errors or require extensive resources (e.g., naive SVD reconstruction), FedASK's two-stage sketching mechanism enables mathematically exact aggregation as formalized in theorem 2.

**Theorem 2** (Aggregate Precision of FedASK). *Let $d_B = dim(span(\bigcup_{k=1}^{K} Range(B_k)))$. If the random projection matrix $\Omega \in \mathbb{R}^{n \times (r+p)}$ is a standard Gaussian random matrix with over-sketching parameter $p$ satisfying $p \geq d_B - r + 2$, then FedASK guarantees that the global update before truncation $\Delta W^t = B^t A^t$ equals the exact average of local updates $\Delta \bar{W} = \frac{1}{K} \sum_{k=1}^{K} B_k A_k$, i.e.,*

$$\|\Delta W^t - \Delta \bar{W}\|_F = 0,$$

*where $\|\cdot\|_F$ denotes the Frobenius norm.*

Theorem 2 provides a theoretical guarantee: the global LoRA update ($\Delta W^t$) computed by FedASK precisely matches the true average of all participating clients' local updates ($\Delta \bar{W}$), provided the specified condition for the over-sketching parameter $p$ is met. Our empirical evaluations (detailed in Section 5.4) suggest that this condition for exact, or near-exact, aggregation is often satisfied with a relatively small value for $p$. This implies that the theoretical precision highlighted by the theorem is practically achievable with minimal over-sketching overhead. Thus, FedASK delivers significant efficiency gains through its projection-based communication without compromising aggregation fidelity in practical settings.

## 5 Experiment

To rigorously evaluate our proposed method, we conduct a comprehensive set of experiments across diverse tasks and models. For natural language processing and mathematical reasoning, we utilize two large-scale Llama-2 models: the 7B and 13B versions [35]. The Llama-2-7B model is fine-tuned on the dolly-15K dataset [44] and assessed on general language understanding benchmarks, including MMLU [17], DROP [7], and HumanEval [5]. Concurrently, the Llama-2-13B model undergoes further fine-tuning with Chain-of-Thought (CoT) prompting [39] on the MetaMathQA dataset [42], with its mathematical reasoning capabilities evaluated using the GSM8K [6], GSM8K-hard, and MATH [18] benchmarks. To demonstrate the versatility of our approach, FedASK, beyond NLP and standard Supervised Fine-Tuning (SFT), we also conduct experiments with Vision Language Models and Reinforcement Learning from Human Feedback (RLHF). In this setting, we perform Direct Preference Optimization (DPO) on Llava-1.5-7b using the SPA-VL safety preference alignment dataset [47]. The evaluation for this task involves MM-SafetyBench [27] to measure resilience to jailbreak attacks via an Attack Success Rate; SIUO [36] to assess safety in cross-modal reasoning;

and BeaverTails-V [21] to provide separate win-rates for harmlessness and helpfulness. Across all experiments, conditions are systematically varied to encompass different privacy budget levels (Section 5.1), degrees of data heterogeneity (Section 5.2), and system robustness (Section 5.4). All evaluations are performed on NVIDIA Tesla A100 GPUs, utilizing half-precision to maximize computational efficiency.

**Baselines.** We compare FedASK with five baseline methods: (1) **FedAvg [30]**: Clients perform local SGD, and the server applies weighted parameter averaging. (2) **FFA-LoRA [34]**: Clients train one low-rank matrix locally, mitigating noise accumulation. (3) **FedSA-LoRA [13]**: Clients locally update both LoRA matrices, with one matrix being transmitted. (4) **FedProx [26]**: Introduces a proximal term in the local client loss to mitigate the effects of data heterogeneity. (5) **Scaffold [33]**: Employs control variates to correct client-server gradient disparities, reducing client drift.

## 5.1 Model Performance with Differential Privacy Guarantees

Experiments with Llama-2-7B on homogeneous data use standardized settings (e.g., $\mathcal{B} = 8$, 10 local steps, 400 rounds) and common LoRA configurations ($r = 64$, $\alpha = 128$), detailed in the appendix. We perform a grid search for learning rates and explore DP budgets $\epsilon \in \{1, 3, 6\}$, following [34]. Results are in Table 2. For Llama-2-13B, we adjust settings to $\mathcal{B} = 6$, 800 total rounds and $r = 128$, with other hyperparameters unchanged. Results are in Table 3.

The introduction of privacy-preserving mechanisms leads to performance degradation in several baseline methods. In contrast, our FedASK algorithm consistently outperforms these methods, regardless of whether privacy protection is enabled, demonstrating its superior generalization under privacy constraints. Interestingly, we find that under certain conditions, adding DP noise improves performance compared to the noiseless case. We attribute this to DP noise serving as implicit regularization, thereby enhancing model robustness.

Table 2: Performance Comparison of Different Algorithms with DP Budgets on Llama-2-7B.

| Task | Priv. Budget | FedASK | FedAvg | FFA-LoRA | FedSA-LoRA | FedProx | Scaffold |
|------|-------------|--------|--------|----------|------------|---------|----------|
| MMLU | Non-Private | **46.15** | 45.13 | 45.98 | 45.19 | 44.98 | 45.65 |
| | $\epsilon = 1$ | **45.80** | 42.07 | 42.76 | 42.9 | 41.99 | 43.41 |
| | $\epsilon = 3$ | **46.25** | 41.49 | 42.72 | 41.13 | 43.17 | 42.47 |
| | $\epsilon = 6$ | **45.78** | 43.34 | 42.82 | 42.84 | 43.70 | 43.80 |
| DROP | Non-Private | **32.09** | 30.2 | 31.34 | 31.23 | 30.99 | 30.01 |
| | $\epsilon = 1$ | **31.23** | 29.55 | 29.10 | 31.04 | 29.51 | 29.66 |
| | $\epsilon = 3$ | **32.08** | 29.26 | 28.40 | 29.40 | 28.50 | 28.75 |
| | $\epsilon = 6$ | **31.36** | 29.30 | 29.40 | 29.26 | 27.57 | 30.20 |
| Human-Eval | Non-Private | **15.24** | 11.59 | 14.02 | 12.2 | 12.2 | 14.63 |
| | $\epsilon = 1$ | **15.24** | 12.80 | 12.20 | 13.41 | 12.20 | 9.76 |
| | $\epsilon = 3$ | **15.24** | 10.37 | 10.98 | 10.98 | 13.41 | 11.59 |
| | $\epsilon = 6$ | **15.85** | 11.59 | 12.20 | 12.80 | 10.98 | 12.20 |

Table 3: Performance Comparison of Different Algorithms with DP Budgets on Llama-2-13B.

| Task | Priv. Budget | FedASK | FedAvg | FFA-LoRA | FedSA-LoRA | FedProx | Scaffold |
|------|-------------|--------|--------|----------|------------|---------|----------|
| GSM8K | Non-Private | **50.0** | 48.5 | 48.4 | 47.2 | 47.8 | 45.6 |
| | $\epsilon = 1$ | **22.7** | 15.5 | 14.2 | 12.2 | 15.2 | 16.1 |
| | $\epsilon = 3$ | **24.8** | 16.5 | 20.0 | 20.2 | 18.0 | 15.8 |
| | $\epsilon = 6$ | **27.7** | 19.3 | 20.2 | 17.3 | 20.1 | 20.3 |
| GSM8K$_{hard}$ | Non-Private | **28.7** | 25.8 | 23.2 | 23.4 | 26.1 | 21.8 |
| | $\epsilon = 1$ | **13.0** | 8.8 | 8.0 | 6.6 | 7.2 | 9.1 |
| | $\epsilon = 3$ | **12.6** | 11.1 | 10.5 | 11.3 | 10.8 | 11.0 |
| | $\epsilon = 6$ | **16.9** | 10.5 | 9.2 | 10.2 | 10.9 | 10.8 |
| Math | Non-Private | **11.8** | 10.3 | 10.8 | 10.7 | 11.7 | 9.8 |
| | $\epsilon = 1$ | **6.9** | 5.2 | 5.8 | 5.6 | 5.6 | 5.8 |
| | $\epsilon = 3$ | **6.6** | 6.1 | 6.0 | 5.9 | 6.4 | 5.5 |
| | $\epsilon = 6$ | **7.6** | 6.2 | 6.0 | 5.9 | 6.7 | 7.1 |

## 5.2 Model Performance with Data Heterogeneity

To evaluate data heterogeneity's impact, we experiment with IID and three non-IID scenarios, using 10 clients with 2 selected per round. In IID settings, data is randomly partitioned. For non-IID settings, we use Dirichlet distribution $\text{Dir}(\alpha)$ with $\alpha \in \{0.1, 0.5, 1.0\}$, following prior work [13].

To evaluate robustness under DP in these non-IID environments, we set the privacy budget $\epsilon = 3$, a common value for balancing privacy and utility. Other settings follow Section 5.1. The experimental results in Table 4 demonstrate FedASK's prominent performance. Across all tasks and data distributions (IID and non-IID), FedASK consistently outperforms baselines, underscoring its effectiveness and robustness. This superior performance stems from FedASK's simultaneous local updates and global aggregation of both A and B matrices. While previous research [13] suggests that matrix A learns global information and matrix B captures local specifics, FedASK's design enables a continuous interaction and fusion of these distinct knowledge types. This inherent information fusion capability enhances the model's adaptation to heterogeneous data distributions.

Table 4: Performance comparison across different data distributions on various tasks.

| Task | Data Dist. | FedASK | FedAvg | FFA-LoRA | FedSA-LoRA | FedProx | Scaffold |
|------|-----------|--------|--------|----------|------------|---------|----------|
| MMLU | IID | **46.25** | 41.49 | 42.72 | 41.13 | 43.17 | 42.47 |
| | Dir(0.1) | **46.04** | 42.69 | 42.54 | 44.27 | 42.61 | 43.05 |
| | Dir(0.5) | **45.95** | 42.11 | 41.46 | 42.72 | 42.98 | 41.97 |
| | Dir(1) | **46.01** | 42.96 | 43.23 | 41.04 | 42.98 | 41.71 |
| DROP | IID | **32.08** | 29.44 | 28.40 | 29.40 | 28.50 | 28.75 |
| | Dir(0.1) | **31.01** | 28.34 | 30.10 | 28.58 | 28.18 | 28.27 |
| | Dir(0.5) | **31.15** | 29.18 | 29.26 | 27.83 | 28.23 | 29.45 |
| | Dir(1) | **31.58** | 30.02 | 28.93 | 29.29 | 29.82 | 30.52 |
| Human-Eval | IID | **15.24** | 10.37 | 10.98 | 10.98 | 13.41 | 11.59 |
| | Dir(0.1) | **13.41** | 7.32 | 10.98 | 12.8 | **13.41** | 7.93 |
| | Dir(0.5) | **14.63** | 10.98 | 13.41 | 12.8 | **14.63** | 9.76 |
| | Dir(1) | **14.02** | 9.76 | 10.98 | 10.37 | 10.37 | 9.15 |

## 5.3 Model Performance with VLMs task and RLHF

To evaluate the efficacy of our method on vision-language tasks and preference alignment, we conduct experiments using the Llava-1.5-7B model. We employ DPO, a form of RLHF, to align the model with safety preferences using the SPA-VL dataset. The FL setup involves 10 clients (IID), 2 selected per round for 600 rounds, and 10 local steps. We apply LoRA to `q_proj`, `v_proj`, and `mm_projector` modules, freezing the `vision_tower`.

The results in Table 5 demonstrate FedASK's superior performance in the VLM alignment task under privacy constraints. At $\epsilon = 6$, FedASK consistently outperforms all baselines, achieving the highest helpfulness (56.36) and harmlessness (65.36) on BeaverTails-V, the lowest ASR (43.40) on MMsafety, and leading scores on SIUO. This advantage stems from a fundamentally more expressive learning process. FedASK avoids the representational bottleneck of fixed-matrix methods, which is particularly limiting in complex, multi-objective tasks like preference alignment. The global SVD reconstruction acts as a powerful information distillation step, re-decomposing the aggregated private updates into a new, more optimal shared basis for both A and B. Furthermore, the inherent DP noise serves as an implicit regularizer, preventing the model from overfitting to specific client preference data and promoting a more generalized alignment. This synergy preserves the model's full expressive capacity while enhancing robustness, explaining its significant margin over baselines even under stricter privacy budgets such as $\epsilon = 1$.

## 5.4 Effect of the Sketching Dimension

Theorem 2 guarantees that FedASK achieves precise global aggregation provided the over-sketching parameter $p$ meets a specified condition, our empirical evaluations aim to demonstrate that this precision is practically attainable with minimal $p$, underscoring FedASK's efficiency. To this end, we conduct two sets of experiments using the Llama-2-7B model and identical settings from Section 5.1 to assess $p$'s impact on both downstream task performance and direct aggregation fidelity.

Table 5: Performance comparison of DPO with LoRA across different privacy budgets.

| Algorithm | Beavertails-V | | | | MMsafety | | SIUO | | | |
| | Helpfulness | | Harmlessness | | ASR ($\downarrow$) | | Effectiveness | | Safety | |
| | $\epsilon$=6 | $\epsilon$=1 | $\epsilon$=6 | $\epsilon$=1 | $\epsilon$=6 | $\epsilon$=1 | $\epsilon$=6 | $\epsilon$=1 | $\epsilon$=6 | $\epsilon$=1 |
|---|---|---|---|---|---|---|---|---|---|---|
| **FedASK** | **56.36** | **53.65** | **65.36** | **62.98** | **43.40** | **46.93** | **87.42** | **85.62** | **31.14** | **28.74** |
| FedAvg | 51.53 | 45.50 | 53.66 | 48.89 | 48.24 | 50.80 | 83.23 | 80.23 | 26.94 | 25.14 |
| FFA-LoRA | 54.66 | 52.38 | 63.32 | 61.96 | 44.39 | 48.45 | 86.23 | 83.83 | 29.34 | 27.54 |
| Scaffold | 51.02 | 48.97 | 52.55 | 46.67 | 50.80 | 51.43 | 83.83 | 76.04 | 27.54 | 23.95 |

First, to evaluate downstream task performance, we test FedASK with varying over-sketching parameters $p \in \{0, 32, 64, 96, 128\}$ on the MMLU benchmark. This is performed under a challenging Non-IID setting ($\alpha = 0.1$) with differential privacy ($\epsilon = 3$). Second, to directly assess aggregation fidelity, we compare FedASK configured with no over-sketching ($p = 0$) against FedAvg. This latter experiment is conducted without DP, across varying numbers of selected clients ($K_s \in \{2, 5, 10, 15, 20\}$) and data heterogeneity levels. After 30 local update rounds by each client, we measure aggregation fidelity using the cosine similarity between FedASK's reconstructed global LoRA update ($\Delta W^t = B^t A^t$) and the ideal average of all the local LoRA updates ($\Delta \bar{W}^t = \frac{1}{K_s} \sum_{k \in K_s} B_k^t A_k^t$).

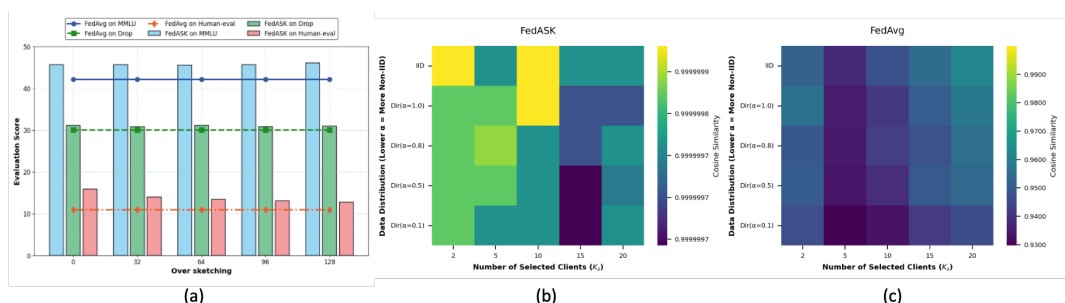

Figure 3: Impact of over-sketching $p$ on FedASK. (a) MMLU score of FedASK versus $p$. (b) Aggregation fidelity of FedASK (with $p = 0$) and (c) FedAvg, measured by cosine similarity with the ideal mean of local updates (1.0 indicates perfect fidelity). Subplots (b) and (c) vary selected clients ($K_s$) and Non-IID degrees.

The empirical results, presented in Figure 3, illustrate that the over-sketching parameter $p$ has a minimal influence on the effectiveness of FedASK. Firstly, (a) shows that FedASK's MMLU performance remains remarkably stable across the tested $p$ values (ranging from 45.63 at $p = 0$ to 46.06 at $p = 128$), even under conditions of significant data heterogeneity and differential privacy. This stability demonstrates a low sensitivity to $p$ to achieve robust downstream utility. Secondly, FedASK's inherent design ensures exceptional aggregation fidelity, even with zero over-sketching ($p = 0$). As shown in (b), FedASK consistently achieves near-perfect cosine similarities (with an error on the order of $10^{-7}$) across a variety of selected clients ($K_s$) and Non-IID degrees. This performance consistently outperforms FedAvg, presented in (c), which typically ranges from 0.92 to 0.96. These findings affirm FedASK's practical efficiency and the attainability of its theoretically guaranteed precise aggregation with negligible over-sketching overhead.

## 6 Conclusion

This paper introduces FedASK, a novel federated LoRA framework that successfully resolves the trade-off between noise amplification and learnability in private federated fine-tuning. By employing a two-stage sketching pipeline, FedASK enables the differentially private and effective update of both LoRA adapters. Our theoretical analysis and comprehensive experiments demonstrate FedASK's superior performance in achieving precise aggregation and robust downstream utility with strong privacy guarantees and practical efficiency. Future studies will focus on extending its applicability to a broader range of model architectures and complex training tasks, as well as exploring further refinements to its privacy-utility trade-off.

## Acknowledgments and Disclosure of Funding

This work was supported by the Natural Science Foundation of China under Grant 62472103 and the National Key Research and Development Program of China under Contract 2024YFA1610900.

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

# A  Theoretical Proof

## A.1  Proof of Lemma 1

*Proof.* Let $A_t \in \mathbb{R}^{r \times d_l}$ and $B_t \in \mathbb{R}^{d_l \times r}$ be the LoRA matrices. The noisy updates under DP-SGD with learning rate $\eta$ are,

$$A_{t+1} = A_t - \eta(\nabla A_t + \xi_A), \tag{9}$$
$$B_{t+1} = B_t - \eta(\nabla B_t + \xi_B), \tag{10}$$

where $\xi_A, \xi_B$ are independent Gaussian noise matrices with i.i.d. entries $\mathcal{N}(0, \sigma_{eff}^2)$, and $\sigma_{eff}^2 = \sigma^2 C^2 / B_{size}^2$. The noise component in the LoRA update $\Delta W = B_{t+1} A_{t+1} - B_t A_t$ is found by substituting (9) and (10),

$$\Delta W = (B_t - \eta(\nabla B_t + \xi_B))(A_t - \eta(\nabla A_t + \xi_A)) - B_t A_t,$$
$$= -\eta(B_t \nabla A_t + \nabla B_t A_t) + \eta^2 \nabla B_t \nabla A_t \quad \text{(Signal part)}$$
$$- \eta(B_t \xi_A + \xi_B A_t) + \eta^2(\nabla B_t \xi_A + \xi_B \nabla A_t + \xi_B \xi_A) \quad \text{(Noise part)}.$$

Thus, $\Delta W_{noise} = X + Y$, where,

$$X = -\eta(B_t \xi_A + \xi_B A_t), \tag{11}$$
$$Y = \eta^2(\nabla B_t \xi_A + \xi_B \nabla A_t + \xi_B \xi_A). \tag{12}$$

The expected noise power is $P_N = \mathbb{E}[||X + Y||_F^2] = \mathbb{E}[||X||_F^2] + \mathbb{E}[||Y||_F^2] + 2\mathbb{E}[\langle X, Y \rangle_F]$.

For the linear noise term power $\mathbb{E}[||X||_F^2]$, using (11),

$$\mathbb{E}[||X||_F^2] = \eta^2 \mathbb{E}[||B_t \xi_A + \xi_B A_t||_F^2],$$
$$= \eta^2 \left( \mathbb{E}[||B_t \xi_A||_F^2] + \mathbb{E}[||\xi_B A_t||_F^2] \right),$$

due to $\mathbb{E}[\langle B_t \xi_A, \xi_B A_t \rangle_F] = 0$ from noise independence and zero mean. We have $\mathbb{E}[\xi_A \xi_A^T] = d_l \sigma_{eff}^2 I_r$ and $\mathbb{E}[\xi_B^T \xi_B] = d_l \sigma_{eff}^2 I_r$. So,

$$\mathbb{E}[||B_t \xi_A||_F^2] = \text{Tr}(B_t \mathbb{E}[\xi_A \xi_A^T] B_t^T) = d_l \sigma_{eff}^2 ||B_t||_F^2,$$
$$\mathbb{E}[||\xi_B A_t||_F^2] = \text{Tr}(A_t A_t^T \mathbb{E}[\xi_B^T \xi_B]) = d_l \sigma_{eff}^2 ||A_t||_F^2.$$

This yields,

$$\mathbb{E}[||X||_F^2] = \eta^2 d_l \sigma_{eff}^2 (||A_t||_F^2 + ||B_t||_F^2). \tag{13}$$

For the cross term $2\mathbb{E}[\langle X, Y \rangle_F]$, using (11) and (12),

$$\mathbb{E}[\langle X, Y \rangle_F] = -\eta^3 \mathbb{E}[\langle B_t \xi_A + \xi_B A_t, \nabla B_t \xi_A + \xi_B \nabla A_t + \xi_B \xi_A \rangle_F].$$

Non-zero expectations arise from,

$$\mathbb{E}[\langle B_t \xi_A, \nabla B_t \xi_A \rangle_F] = \text{Tr}(B_t^T \nabla B_t \mathbb{E}[\xi_A \xi_A^T]) = d_l \sigma_{eff}^2 \text{Tr}(B_t^T \nabla B_t),$$
$$\mathbb{E}[\langle \xi_B A_t, \xi_B \nabla A_t \rangle_F] = \text{Tr}(A_t^T \mathbb{E}[\xi_B^T \xi_B] \nabla A_t) = d_l \sigma_{eff}^2 \text{Tr}(A_t^T \nabla A_t).$$

Other terms are zero since the independence of noise. Thus,

$$2\mathbb{E}[\langle X, Y \rangle_F] = -2\eta^3 d_l \sigma_{eff}^2 (\text{Tr}(B_t^T \nabla B_t) + \text{Tr}(A_t^T \nabla A_t)). \tag{14}$$

For the higher-order noise term power $\mathbb{E}[||Y||_F^2]$, using (12), the dominant component is,

$$\mathbb{E}[||\eta^2 \xi_B \xi_A||_F^2] = \eta^4 \mathbb{E}[\text{Tr}(\xi_B \xi_A \xi_A^T \xi_B^T)], \tag{15}$$
$$= \eta^4 \text{Tr}(\mathbb{E}[\xi_B \mathbb{E}[\xi_A \xi_A^T] \xi_B^T]), \tag{16}$$
$$= \eta^4 \text{Tr}(\mathbb{E}[\xi_B (d_l \sigma_{eff}^2 I_r) \xi_B^T]), \tag{17}$$
$$= \eta^4 d_l \sigma_{eff}^2 \mathbb{E}[||\xi_B||_F^2] = \eta^4 d_l^2 r (\sigma_{eff}^2)^2. \tag{18}$$

Other terms in $\mathbb{E}[||Y||_F^2]$ are $O(\eta^4 \sigma_{eff}^2 d_l r(||\nabla A_t||_F^2 + ||\nabla B_t||_F^2))$.

Combining the dominant terms from (13) and the dominant part of $\mathbb{E}[\|Y\|_F^2]$ (derived in (18)), and noting the cross-term (14) is often less dominant, we approximate $P_N$. Substituting $\sigma_{eff}^2 = \sigma^2 C^2 / B_{size}^2$, we get,

$$\mathbb{E}[\|\Delta W_{noise}\|_F^2] \approx \eta^2 \frac{\sigma^2 C^2}{B_{size}^2} d_l r(\|A_t\|_F^2 + \|B_t\|_F^2) + \eta^4 \frac{\sigma^4 C^4}{B_{size}^4} d_l^2 r, \tag{19}$$

which matches the expression stated in Lemma 1, highlighting the linear and dominant quadratic noise terms. $\square$

## A.2 Proof of Theorem 1

To facilitate the privacy analysis of FedASK, we first recall several fundamental concepts and properties of differential privacy. We leverage the advanced composition theorem and subsampling theorem for $(\epsilon, \delta)$-differential privacy:

**Lemma 2** (Advanced Composition [10]). *Let $M_1, \ldots, M_k$ be a sequence of $k$ adaptive mechanisms, where each $M_i$ provides $(\epsilon, \delta)$-differential privacy. For any $\delta' > 0$, the composed mechanism $M = (M_1, \ldots, M_k)$ is $(\epsilon_{total}, k\delta + \delta')$-differentially private, where*

$$\epsilon_{total} = \sqrt{2k \ln(1/\delta')}\epsilon + k\epsilon(e^\epsilon - 1).$$

*If $\epsilon \ll 1$, then for any $\delta' > 0$, the composed mechanism $M$ is $(\epsilon'_{total}, k\delta + \delta')$-differentially private, where $\epsilon'_{total} \approx \sqrt{2k \ln(1/\delta')}\epsilon + k\epsilon^2$.*

**Lemma 3** (Privacy Amplification by Subsampling [37]). *Let $\mathcal{M}$ be an $(\epsilon, \delta)$-differentially private mechanism. If $\mathcal{M}$ is run on a random sample of size $m$ drawn uniformly without replacement from a dataset of size $N$ (where $m \leq N$), let $\gamma = m/N$ be the sampling ratio. Then, the subsampled mechanism $\mathcal{M}_{subsample}$ is $(\epsilon', \delta')$-differentially private, where,*

$$\epsilon' = \log(1 + \gamma(e^\epsilon - 1)),$$
$$\delta' = \gamma\delta.$$

While $(\epsilon, \delta)$-DP provides a worst-case privacy guarantee, analyzing the precise privacy loss under composition, especially for Gaussian mechanisms, can be complex. Rényi Differential Privacy (RDP) [31] offers a convenient framework for tracking privacy loss, providing tighter bounds under composition. We define RDP as follows:

**Definition 2** (RDP [31]). *A randomized mechanism $\mathcal{M} : \mathcal{D} \rightarrow \mathcal{R}$ satisfies $(\alpha, R)$-Rényi Differential Privacy (RDP) if for any neighboring datasets $D, D' \in \mathcal{D}$ (differing in one individual's data), and for all $\alpha > 1$,*
$$D_\alpha(\mathcal{M}(D)\|\mathcal{M}(D')) \leq R,$$
*where $D_\alpha(P\|Q) = \frac{1}{\alpha-1} \ln \mathbb{E}_{x \sim Q(x)} \left[ \left( \frac{P(x)}{Q(x)} \right)^\alpha \right]$ is the Rényi divergence of order $\alpha$.*

RDP possesses several useful properties that simplify privacy analysis.

**Lemma 4** (Post-processing of RDP [31]). *Let $\mathcal{M} : \mathcal{D} \rightarrow \mathcal{R}$ be a mechanism that satisfies $(\alpha, R)$-RDP. Let $g : \mathcal{R} \rightarrow \mathcal{R}'$ be an arbitrary randomized mapping (a post-processing function). Then the mechanism $g \circ \mathcal{M} : \mathcal{D} \rightarrow \mathcal{R}'$ also satisfies $(\alpha, R)$-RDP.*

**Lemma 5** (Adaptive Sequential Composition of RDP [31]). *Let $\mathcal{M}_1 : \mathcal{D} \rightarrow \mathcal{R}_1$ be a mechanism satisfying $(\alpha, R_1)$-RDP. Let $\mathcal{M}_2 : \mathcal{R}_1 \times \mathcal{D} \rightarrow \mathcal{R}_2$ be a mechanism such that for any fixed output $o_1 \in \mathcal{R}_1$ of $\mathcal{M}_1$, $\mathcal{M}_2(o_1, \cdot)$ satisfies $(\alpha, R_2)$-RDP. Then the mechanism $\mathcal{M}(D) = (\mathcal{M}_1(D), \mathcal{M}_2(\mathcal{M}_1(D), D))$, which outputs the pair $(o_1, o_2)$ where $o_1 \sim \mathcal{M}_1(D)$ and $o_2 \sim \mathcal{M}_2(o_1, D)$, satisfies $(\alpha, R_1 + R_2)$-RDP.*

**Lemma 6** (Conversion from RDP to $(\epsilon, \delta)$-DP [4]). *If a randomized mechanism $\mathcal{M}$ satisfies $(\alpha, R)$-RDP, then it satisfies $(R + \ln((\alpha - 1)/\alpha) - (\ln \delta + \ln \alpha)/(\alpha - 1), \delta)$-DP for any $0 < \delta < 1$.*

The core mechanism for achieving privacy in FedASK involves adding Gaussian noise. The RDP of a (subsampled) Gaussian mechanism is characterized as follows:

**Lemma 7** (Approximate RDP for $q_{\mathcal{D}}$-Subsampled Gaussian Mechanism [32])**.** *Consider a $q_{\mathcal{D}}$-subsampled Gaussian mechanism with noise variance $\sigma_g^2$. Under the assumption that the data subsampling ratio $q_{\mathcal{D}}$ is small (i.e., $q_{\mathcal{D}} = o(1)$), and assuming the mechanism operates in a high privacy regime (Assumption 1-(iii) in [32]), for any real number $\alpha > 1$, the mechanism satisfies $(\alpha, R')$-RDP, where $R' = \mathcal{O}(\frac{q_{\mathcal{D}}^2(\alpha+1)}{\sigma_g^2})$.*

*Proof.* The proof proceeds in several steps. We first analyze the RDP guarantee for the information transmitted by a single client $k$ in one communication round $t$ with respect to its local dataset $\mathcal{D}_k$. Then, we analyze the RDP of the aggregated global update at the server. Finally, we compose the privacy loss over $T$ communication rounds, incorporate the amplification due to user subsampling, and convert the total RDP to an $(\epsilon, \delta)$-DP guarantee.

**Step 1: RDP of Local Updates and Transmitted Sketches by Client $k$ w.r.t. $\mathcal{D}_k$.**

Privacy guarantee for first sketching Client $k$ executes LocalUpdateDP (using $m$ local steps, data subsampling ratio $s$, fixed $A_k^t = A^{t-1}$, and noise parameter $\sigma^2$) on its private dataset $\mathcal{D}_k$ to compute the local LoRA matrix $B_k^t$. By Lemma 5 and Lemma 7, $B_k^t$ satisfies $(\alpha, \mathcal{O}(\frac{mq_{\mathcal{D}}^2(\alpha+1)}{\sigma^2}))$-RDP w.r.t. $\mathcal{D}_k$. The first sketch, $Y_k^{\text{proj}} = B_k^t(A_k^t\Omega)$, is derived from $B_k^t$ by post-processing (Lemma 4), since $A_k^t$ and $\Omega$ are independent of $\mathcal{D}_k$ in this context. Thus, $Y_k^{\text{proj}}$ also satisfies $(\alpha, R_k^{(t)}(\alpha))$-RDP w.r.t. $\mathcal{D}_k$, where $R_k^{(t)}(\alpha) = \mathcal{O}(\frac{mq_{\mathcal{D}}^2(\alpha+1)}{\sigma^2})$.

Privacy guarantee for second sketching: After the client $k$ sends $Y_k^{proj}$ to the server, the server computes an orthonormal basis $Q^t = \text{QR}(\sum_{j \in \mathcal{K}_t} Y_j^{\text{proj}})$ and broadcasts $Q^t$ back to the participating clients, including the client $k$. The client $k$ then computes its second sketch $\tilde{Y}_k^{proj} = (A_k^t)^T((B_k^t)^T Q^t)$. The calculation of $\tilde{Y}_k^{proj}$ utilizes the already privatized matrix $B_k^t$, the public matrix $A_k^t$, and the received matrix $Q^t$. For client $k$, $Q^t$ is an external input provided by the server, and this computation does not involve fresh access to its private dataset $\mathcal{D}_k$. Consequently, according to Lemma 6, $\tilde{Y}_k^{proj}$ also satisfies $(\alpha, R_k^{(t)}(\alpha))$-RDP with respect to $\mathcal{D}_k$.

So we could have that, for the local uplink transmitted information, $Y_j^{\text{proj}}$ and $\tilde{Y}_k^{proj}$ satisfy the $(\alpha, R_k^{(t)}(\alpha))$-RDP w.r.t. $\mathcal{D}_k$, where

$$R_k^{(t)}(\alpha) = \mathcal{O}(\frac{mq_{\mathcal{D}}^2(\alpha+1)}{\sigma^2}). \tag{20}$$

**Step 2: DP guarantee of $(A^t, B^t)$ w.r.t. the Joint Dataset $\mathcal{D} = \bigcup_k \mathcal{D}_k$.**

The server aggregates the second sketches to form $\tilde{Y}_{\text{agg}}^t = \sum_{k \in \mathcal{K}_t} \tilde{Y}_k^{\text{proj}}$. Based on the $(\alpha, \epsilon_k^{(t)}(\alpha))$-RDP guarantee for each client's transmitted information w.r.t. its local data (Step 1) and the disjointness of client datasets, the mechanism producing an appropriately scaled aggregate over the $K_s = |\mathcal{K}_t|$ participating clients yields $(\alpha, R_{\text{agg}}^{(t)}(\alpha))$-RDP for $\tilde{Y}_{\text{agg}}^t$ w.r.t. the joint data $D_{\mathcal{K}_t} = \bigcup_{k \in \mathcal{K}_t} \mathcal{D}_k$, where

$$R_{\text{agg}}^{(t)}(\alpha) = \epsilon_k^{(t)}(\alpha)/K_s = \mathcal{O}\left(\frac{mq_{\mathcal{D}}^2(\alpha+1)}{K_s\sigma^2}\right). \tag{21}$$

Since the global LoRA matrices $(A^t, B^t)$ are derived via SVD of $\tilde{Y}_{\text{agg}}^t$ (a post-processing step, Lemma 4), they inherit this RDP guarantee. Consequently, by applying the RDP to DP conversion (Lemma 6), for any $0 < \delta_0 < 1$, $(A^t, B^t)$ satisfy $(\epsilon_0^{(t)}, \delta_0)$-DP w.r.t. $D_{\mathcal{K}_t}$, with $\epsilon_0^{(t)}$ given by:

$$\epsilon_0^{(t)}(\alpha, \delta_0) = R_{\text{agg}}^{(t)}(\alpha) + \ln(\frac{\alpha-1}{\alpha}) - \frac{\ln\delta_0 + \ln\alpha}{\alpha-1}. \tag{22}$$

**Step 3: DP guarantee of $(A^T, B^T)$ w.r.t. the Joint Dataset $\mathcal{D} = \bigcup_k \mathcal{D}_k$.**

The per-round guarantee $(\epsilon_0^{(t)}, \delta_0)$ w.r.t. $D_{\mathcal{K}_t}$ is amplified by client subsampling (ratio $q_{\mathcal{K}}$ from $K$ total clients) using Lemma 3, yielding an $(q_{\mathcal{K}}\epsilon_0^{(t)}(\alpha, \delta_0), q_{\mathcal{K}}\delta_0)$-DP guarantee per round w.r.t. the full dataset $D = \bigcup_{i=1}^K \mathcal{D}_i$.

Composing this $(q_\mathcal{K}\epsilon_0^{(t)}(\alpha, \delta_0), q_\mathcal{K}\delta_0)$-DP mechanism over $T$ adaptive communication rounds using Lemma 2 , for a chosen $\delta_1 > 0$, gives the final $(\epsilon, \delta)$-DP guarantee for $(A^T, B^T)$:

$$\epsilon = q_\mathcal{K}\sqrt{2T\ln(1/\delta_1)}\mathcal{O}\left(\frac{mq_\mathcal{D}^2(\alpha+1)}{K_s\sigma^2} + \ln(\frac{\alpha-1}{\alpha}) - \frac{\ln\delta_0 + \ln\alpha}{\alpha-1}\right), \tag{23}$$

$$\delta = q_\mathcal{K}T\delta_0 + \delta_1. \tag{24}$$

The specific noise variance $\sigma^2$ required in Theorem 1 is derived by appropriately setting $\epsilon_0^{(t)}, \delta_0$ (based on RDP conversion from Step 2) and $\delta_1$ to meet the target overall $(\epsilon, \delta)$, then solving for $\sigma^2$.

**Step 4: Analysis of the RDP Order $\alpha$.**

Let $\delta_0$ and $\delta_1$ be $\delta/2$ and $\delta/(2q_\mathcal{K}T)$ respectively, we now trying to acquire an expression of $\alpha$ through the following minization problem

$$\min_{\alpha>1}\frac{mq_\mathcal{D}^2(\alpha+1)}{K_s\sigma^2} + \ln(\frac{\alpha-1}{\alpha}) - \frac{\ln(\delta/2) + \ln\alpha}{\alpha-1}. \tag{25}$$

Let $C_1 = \frac{mq_\mathcal{D}^2}{K_s\sigma^2}$, equation (25) find an optimal $\alpha > 1$ that minimizes $\epsilon$ is approximated by setting the derivative of the dominant terms to zero.

$$C_1(\alpha-1)^2 + \ln(\delta_0\alpha) = 0. \tag{26}$$

Assume $\alpha \gg 1$, then, $(\alpha-1)^2 \approx \alpha^2$ and the term $\ln(\delta_0\alpha)$ can be written as $\ln\delta_0 + \ln\alpha$.

Substituting these approximations into the condition (26), we get:

$$C_1\alpha^2 + \ln\delta_0 + \ln\alpha \approx 0. \tag{27}$$

If $\alpha$ is sufficiently large such that $C_1\alpha^2$ dominates $\ln\alpha$ (i.e., the $\alpha^2$ term grows much faster than $\ln\alpha$), we can further simplify Eq. (27) by neglecting the $\ln\alpha$ term relative to $C_1\alpha^2$ and $\ln\delta_0$ (especially if $|\ln\delta_0|$ is large, which is true for small $\delta_0$). This yields:

$$C_1\alpha^2 \approx -\ln\delta_0. \tag{28}$$

From Eq. (28), we obtain an approximate expression for the optimal $\alpha$:

$$\alpha_{\text{opt}} \approx \sqrt{\frac{-\ln\delta_0}{C_1}}. \tag{29}$$

Substituting $\delta_0 = \delta/(2q_\mathcal{K}T)$ and $C_1 = \frac{mq_\mathcal{D}^2}{K_s\sigma^2}$:

$$\alpha_{\text{opt}} \approx \sqrt{\frac{\ln(2q_\mathcal{K}T/\delta)\cdot K_s\sigma^2}{mq_\mathcal{D}^2}}. \tag{30}$$

Plug (30) into (24), we could conclude that to reach $(\epsilon, \delta)$-DP, FedASK nessesitates a noise of

$$\sigma^2 = \mathcal{O}\left(\frac{q_\mathcal{D}^2\cdot m\cdot q_\mathcal{K}\cdot T\cdot\ln(2/\delta)\cdot\ln(2Tq_\mathcal{K}/\delta)}{\epsilon^2\cdot K}\right). \tag{31}$$

This finishes the proof of theorem 1.

$\square$

## A.3 Proof of Theorem 2

**Lemma 8** (Expected Frobenius Norm Error of Random Projection [15]). *Let $A \in \mathbb{R}^{m\times n}$ be a matrix with singular values $\sigma_1 \geq \sigma_2 \geq \ldots$. Choose a target approximation rank $k_{approx} \geq 1$ and an oversampling number $s_{over} \geq 2$ such that the total number of random projection vectors $l = k_{approx} + s_{over} \leq \min\{m, n\}$. Let $\Omega \in \mathbb{R}^{n\times l}$ be a standard Gaussian random matrix, and let $Y = A\Omega$. Let $P_Y$ be the orthogonal projector onto $Range(Y)$. Then, the expected Frobenius norm of the error in approximating $A$ by its projection $P_Y A$ is bounded by,*

$$\mathbb{E}\|(I - P_Y)A\|_F \leq \left(1 + \frac{k_{approx}}{s_{over}-1}\right)^{1/2}\left(\sum_{j>k_{approx}}\sigma_j^2(A)\right)^{1/2}.$$

*Proof.* Let the average of local updates be defined as:

$$\Delta \bar{W} = \frac{1}{K} \sum_{k=1}^{K} B_k A_k. \tag{32}$$

The first sketching follows as:

$$Y^{proj} = \frac{1}{K} \sum_{k=1}^{K} B_k (A_k \Omega) = \Delta \bar{W} \Omega, \tag{33}$$

where $\Omega \in \mathbb{R}^{n \times (r+p)}$ is standard Gaussian. The dimension of the random subspace is $r + p$.

The second aggregated projection is:

$$\tilde{Y}^{proj} = \frac{1}{K} \sum_{k=1}^{K} (A_k^\top (B_k^\top Q)) = (\Delta \bar{W})^\top Q \tag{34}$$

Let $\text{SVD}(Q^\top \Delta \bar{W}) = U \Sigma V^\top$.

$$Q^\top \Delta \bar{W} = U \Sigma V^\top \tag{35}$$

The global update $\Delta W^t = B^t A^t = (QU\Sigma^{1/2})(\Sigma^{1/2} V^\top) = QU\Sigma V^\top$.

$$\Delta W^t = Q(Q^\top \Delta \bar{W}) = QQ^\top \Delta \bar{W} \tag{36}$$

The condition $p \geq d_B - r + 2$ implies $r + p \geq d_B + 2$. Since $\text{rank}(\Delta \bar{W}) \leq d_B$, we have $r + p + 2 \geq \text{rank}(\Delta \bar{W})$. Apply Lemma 8 with $A = \Delta \bar{W}$, let the target rank be $k_{\text{approx}} = r + p$, and $s_{\text{over}} = 2$. For $k_{\text{approx}} > \text{rank}(\Delta \bar{W})$, the singular values $\sigma_{k_{\text{approx}}+1}$ and beyond are zero. Therefore, we could come to the conclusion of Theorem 2. $\qquad \square$

## B  Future Explorations

While FedASK demonstrates notable strengths, we identify the following areas for future exploration:

- **Local Matrix Update Strategies:** FedASK currently fixes the local $A_k$ matrix during differentially private updates in one communication round. Investigating alternating local updates for both $A_k$ and $B_k$ matrices under differential privacy could reveal different learning dynamics and performance trade-offs.

- **Broader Model Applicability:** Our validation of FedASK is currently limited to Large Language Models. Assessing its efficacy and potential adaptations for other architectures, such as vision transformers or diffusion models, remains an open research direction.

- **Advanced Training Paradigms:** The current study focuses on fine-tuning for standard language and reasoning tasks. Extending FedASK to more complex federated and private training paradigms, such as model alignment (e.g., RLHF), presents a valuable avenue for future work.

## C  More Experimets results

### C.1  Training details

**NLP Task, Model Performance with Privacy Guarantee.** Experiments with the Llama-2-7B model focus on homogeneous data distributions. To ensure fair comparisons, standardized settings include a batch size $\mathcal{B}$ of 8, 10 local update steps, and 4000 total communication rounds. Transformer-related hyperparameters, such as the sequence length $l_{seq}$ of 128, align with previous studies [19]. The LoRA rank $r$ is fixed at 64, with the scaling factor $\alpha$ set to twice the rank. Optimal performance is determined through a grid search over learning rates $\{5e-5, 1e-4, 2e-4, 5e-4\}$, and in differential privacy scenarios, privacy budgets $\epsilon \in \{1, 3, 6\}$ are explored, consistent with prior work [34]. For the Llama-2-13B experiments, we use a batch size $\mathcal{B}$ of 6, extend the total communication

rounds to 8000, adjust the sequence length $l_{seq}$ to 968, and increase the LoRA rank $r$ to 128. Other settings remain the same as in the Llama-2-7B experiments. Results are summarized in Table 3.

**NLP Task, Model Performance with Data Heterogenety.** To evaluate data heterogeneity's impact, we experiment with IID and three non-IID scenarios, using 10 clients with 2 selected per round. In IID settings, data is randomly partitioned. For non-IID settings, we use Dirichlet distribution $Dir(\alpha)$ with $\alpha \in \{0.1, 0.5, 1.0\}$, following prior work [13]. To evaluate robustness under DP in these non-IID environments, we set the privacy budget $\epsilon = 3$, a common value for balancing privacy and utility. Other settings keep the same.

**VLM Task, Model Performance with RLHF.** To evaluate the efficacy of our method on vision-language tasks and preference alignment, we conduct experiments using the Llava-1.5-7B model. We employ DPO, a form of RLHF, to align the model with safety preferences using the SPA-VL dataset. The training dataset is curated by first filtering the source data to include only samples where the "chosen" response exceeds 450 characters, followed by a balanced sampling of 2,888 entries from each top-level category to ensure diversity. The federated learning environment consists of 10 clients with an IID data distribution, where 2 clients are selected per round for a total of 600 communication rounds, with each client performing 10 local update steps. Key hyperparameters include a LoRA rank of 256 applied to the `q_proj`, `v_proj`, and `mm_projector` modules while freezing the `vision_tower`, a batch size of 2, and a learning rate of 1e-5. We evaluate performance under Non-Private and two differential privacy budgets, $\epsilon \in \{, 6\}$.

Table 6: More Performance comparison of algorithms across different privacy budgets.

| Algorithm | Beavertails-V | | | | | | MMsafety | | | SIUO | | | | | |
| | Help | | | Harm | | | ASR ($\downarrow$) | | | Effective | | | Safety | | |
| | No DP | $\epsilon$=6 | $\epsilon$=1 | No DP | $\epsilon$=6 | $\epsilon$=1 | No DP | $\epsilon$=6 | $\epsilon$=1 | No DP | $\epsilon$=6 | $\epsilon$=1 | No DP | $\epsilon$=6 | $\epsilon$=1 |
|---|---|---|---|---|---|---|---|---|---|---|---|---|---|---|---|
| **FedASK** | **55.78** | **56.36** | **53.65** | **65.13** | **65.36** | **62.98** | **43.21** | **43.40** | **46.93** | 88.62 | **87.42** | **85.62** | **30.53** | **31.14** | **28.74** |
| **FedAvg** | 53.98 | 51.53 | 45.50 | 60.61 | 53.66 | 48.89 | 45.63 | 48.24 | 50.80 | **89.82** | 83.23 | 80.23 | 29.94 | 26.94 | 25.14 |
| **FFA-LoRA** | 55.68 | 54.66 | 52.38 | 63.43 | 63.32 | 61.96 | 44.93 | 44.39 | 48.45 | 86.83 | 86.23 | 83.83 | 28.74 | 29.34 | 27.54 |
| **Scaffold** | 52.01 | 51.02 | 48.97 | 57.61 | 52.55 | 46.67 | 48.75 | 50.80 | 51.43 | 88.02 | 83.83 | 76.04 | 28.14 | 27.54 | 23.95 |

## C.2 Error-Bar of Current Experiments

This section presents error bar experiments, reporting the mean $\pm$ standard error of the mean (SEM) over five independent runs, to substantiate the stability and performance of the FedASK framework. These evaluations cover varied differential privacy (DP) budgets and non-IID data distributions for Llama-2-7B and Llama-2-13B models on the dolly-15K and MetaMathQA datasets, respectively. Unless otherwise specified, all other parameters, such as batch size and communication rounds, align with the primary experimental configurations detailed in Section 5.

For these error-bar evaluations, specific learning rates and LoRA ranks were employed. Llama-2-7B experiments (Tables 7 and 9) used LoRA rank $r = 64$. For IID data with varying DP budgets (Table 7), baseline learning rates were $2 \times 10^{-4}$ (non-DP) and $1 \times 10^{-4}$ (DP); FedASK and other LoRA methods used $5 \times 10^{-5}$ (non-DP) and $4 \times 10^{-4}$ (DP) with gradient clipping of 1.0. For non-IID evaluations at DP $\epsilon = 3$ (Table 9), baseline learning rates were $1 \times 10^{-4}$, and LoRA-based methods (including FedASK) used $4 \times 10^{-4}$ with 1.0 gradient clipping. The Llama-2-13B experiments with IID data (Table 8) utilized a LoRA rank $r = 128$; FedASK learning rates were $5 \times 10^{-4}$ (non-DP) and $4 \times 10^{-4}$ (DP), while baselines used $2 \times 10^{-4}$.

The inclusion of mean $\pm$ SEM from five runs in these experiments offers robust statistical validation of the delineated advantages of FedASK, reinforcing the conclusions drawn from single-run experiments in the main paper. As detailed in Table 7 and Table 8 , FedASK outperforms or performs comparable to baseline methods in non-private settings and DP budgets of $\epsilon \in \{1, 3, 6\}$, frequently producing comparable or reduced SEMs, highlighting its capacity to achieve an effective equilibrium between model utility and privacy preservation. This demonstrated robustness is further evident in scenarios characterized by data heterogeneity; Table 9 reveals FedASK's consistent maintenance of leading average performance alongside constrained variability, as indicated by the SEMs, across diverse non-IID Dirichlet distributions ($\alpha \in \{0.1, 0.5, 1.0\}$), corroborating the adaptability observations presented in Section 5.2.

Table 7: Performance Comparison (Mean ± SEM from five runs) on Llama-2-7B with Different DP Budgets.

| Task | Priv. Budget | **FedASK** | FedAvg | FFA-LoRA | FedSA-LoRA | FedProx | Scaffold |
|------|------|------|------|------|------|------|------|
| MMLU | Non-Private | $46.34 \pm 0.19$ | $43.13 \pm 2.01$ | $45.72 \pm 0.27$ | $44.63 \pm 0.57$ | $43.55 \pm 1.44$ | $44.49 \pm 1.17$ |
| | $\epsilon = 1$ | $45.79 \pm 0.01$ | $42.82 \pm 0.75$ | $44.15 \pm 1.39$ | $43.16 \pm 0.25$ | $43.00 \pm 1.01$ | $43.53 \pm 0.12$ |
| | $\epsilon = 3$ | $45.88 \pm 0.37$ | $42.43 \pm 0.94$ | $43.82 \pm 1.10$ | $42.30 \pm 1.17$ | $43.07 \pm 0.11$ | $42.88 \pm 0.41$ |
| | $\epsilon = 6$ | $45.73 \pm 0.05$ | $43.46 \pm 0.12$ | $44.02 \pm 1.20$ | $43.30 \pm 0.54$ | $43.00 \pm 0.71$ | $43.43 \pm 0.37$ |
| DROP | Non-Private | $32.01 \pm 0.08$ | $30.31 \pm 0.11$ | $31.65 \pm 0.31$ | $30.86 \pm 0.37$ | $30.95 \pm 0.04$ | $30.87 \pm 0.86$ |
| | $\epsilon = 1$ | $31.33 \pm 0.10$ | $30.49 \pm 0.94$ | $30.41 \pm 1.31$ | $29.96 \pm 1.08$ | $30.18 \pm 0.67$ | $30.08 \pm 0.42$ |
| | $\epsilon = 3$ | $31.06 \pm 1.02$ | $29.69 \pm 0.43$ | $29.87 \pm 1.47$ | $29.66 \pm 0.26$ | $29.07 \pm 0.57$ | $28.90 \pm 0.15$ |
| | $\epsilon = 6$ | $31.27 \pm 0.10$ | $29.46 \pm 0.15$ | $30.37 \pm 0.97$ | $29.91 \pm 0.65$ | $28.64 \pm 1.07$ | $30.19 \pm 0.01$ |
| Human-Eval | Non-Private | $14.63 \pm 0.61$ | $13.42 \pm 1.83$ | $14.02 \pm 0.02$ | $12.81 \pm 0.61$ | $12.81 \pm 0.61$ | $14.63 \pm 0.00$ |
| | $\epsilon = 1$ | $13.72 \pm 1.52$ | $11.28 \pm 1.52$ | $12.50 \pm 0.30$ | $11.28 \pm 2.13$ | $8.85 \pm 3.35$ | $8.54 \pm 1.22$ |
| | $\epsilon = 3$ | $14.02 \pm 1.22$ | $7.63 \pm 2.74$ | $11.59 \pm 0.61$ | $8.85 \pm 2.13$ | $10.06 \pm 3.35$ | $9.76 \pm 1.83$ |
| | $\epsilon = 6$ | $15.85 \pm 0.02$ | $9.76 \pm 1.83$ | $11.90 \pm 0.31$ | $10.67 \pm 2.13$ | $8.85 \pm 2.13$ | $9.76 \pm 2.44$ |

Table 8: Performance Comparison (Mean ± SEM from five runs) on Llama-2-13B with Different DP Budgets.

| Task | Priv. Budget | **FedASK** | FedAvg | FFA-LoRA | FedSA-LoRA | FedProx | Scaffold |
|------|------|------|------|------|------|------|------|
| GSM8K | Non-Private | $51.40 \pm 1.40$ | $46.25 \pm 2.25$ | $48.50 \pm 0.10$ | $50.00 \pm 2.80$ | $47.95 \pm 0.15$ | $46.95 \pm 1.35$ |
| | $\epsilon = 1$ | $24.95 \pm 2.25$ | $16.40 \pm 0.90$ | $14.25 \pm 0.05$ | $14.50 \pm 2.30$ | $16.00 \pm 0.80$ | $16.45 \pm 0.35$ |
| | $\epsilon = 3$ | $25.35 \pm 0.55$ | $19.60 \pm 3.10$ | $18.60 \pm 1.40$ | $21.20 \pm 1.00$ | $20.20 \pm 2.20$ | $18.30 \pm 2.50$ |
| | $\epsilon = 6$ | $24.35 \pm 3.35$ | $20.05 \pm 0.75$ | $19.15 \pm 0.95$ | $18.45 \pm 1.15$ | $22.05 \pm 1.95$ | $20.35 \pm 0.05$ |
| GSM8K$_{hard}$ | Non-Private | $23.90 \pm 4.80$ | $21.90 \pm 3.90$ | $22.20 \pm 1.00$ | $22.10 \pm 1.30$ | $23.05 \pm 3.05$ | $20.95 \pm 0.85$ |
| | $\epsilon = 1$ | $13.65 \pm 0.65$ | $10.25 \pm 1.45$ | $7.85 \pm 0.15$ | $8.25 \pm 1.65$ | $7.90 \pm 0.70$ | $9.35 \pm 0.25$ |
| | $\epsilon = 3$ | $13.00 \pm 0.40$ | $11.90 \pm 0.80$ | $10.25 \pm 0.25$ | $11.55 \pm 0.25$ | $11.35 \pm 0.55$ | $11.20 \pm 0.20$ |
| | $\epsilon = 6$ | $13.30 \pm 3.60$ | $11.30 \pm 0.80$ | $9.40 \pm 0.20$ | $10.55 \pm 0.35$ | $11.60 \pm 0.70$ | $11.40 \pm 0.60$ |
| Math | Non-Private | $12.55 \pm 0.75$ | $9.30 \pm 1.00$ | $10.25 \pm 0.55$ | $10.60 \pm 0.10$ | $10.90 \pm 0.80$ | $10.00 \pm 0.20$ |
| | $\epsilon = 1$ | $7.25 \pm 0.35$ | $5.55 \pm 0.35$ | $5.50 \pm 0.30$ | $5.85 \pm 0.25$ | $5.85 \pm 0.25$ | $5.55 \pm 0.25$ |
| | $\epsilon = 3$ | $7.20 \pm 0.60$ | $6.50 \pm 0.40$ | $6.20 \pm 0.20$ | $6.15 \pm 0.25$ | $6.30 \pm 0.10$ | $5.95 \pm 0.45$ |
| | $\epsilon = 6$ | $6.50 \pm 1.10$ | $6.35 \pm 0.15$ | $6.15 \pm 0.15$ | $6.05 \pm 0.15$ | $6.50 \pm 0.20$ | $7.00 \pm 0.10$ |

Table 9: Performance Comparison (Mean ± SEM from five runs) for DP Budget $\epsilon = 3$ across Different Data Distributions on Llama-2-7B.

| Task | Data Dist. | **FedASK** | FedAvg | FFA-LoRA | FedSA-LoRA | FedProx | Scaffold |
|------|------|------|------|------|------|------|------|
| MMLU | IID | $45.88 \pm 0.37$ | $42.43 \pm 0.94$ | $43.82 \pm 1.10$ | $42.30 \pm 1.17$ | $43.07 \pm 0.11$ | $42.88 \pm 0.41$ |
| | Dir(0.1) | $45.21 \pm 0.84$ | $42.85 \pm 0.16$ | $43.64 \pm 1.09$ | $44.11 \pm 0.16$ | $42.99 \pm 0.39$ | $42.33 \pm 0.73$ |
| | Dir(0.5) | $45.05 \pm 0.90$ | $42.71 \pm 0.60$ | $43.22 \pm 1.76$ | $43.26 \pm 0.54$ | $42.66 \pm 0.32$ | $42.91 \pm 0.94$ |
| | Dir(1) | $45.26 \pm 0.75$ | $42.75 \pm 0.21$ | $44.60 \pm 1.37$ | $42.37 \pm 1.33$ | $43.10 \pm 0.12$ | $42.69 \pm 0.98$ |
| DROP | IID | $31.10 \pm 1.04$ | $29.78 \pm 0.34$ | $29.87 \pm 1.47$ | $29.66 \pm 0.26$ | $29.07 \pm 0.57$ | $28.90 \pm 0.15$ |
| | Dir(0.1) | $30.85 \pm 0.17$ | $29.27 \pm 0.93$ | $30.52 \pm 0.42$ | $28.72 \pm 0.14$ | $28.53 \pm 0.35$ | $28.29 \pm 0.02$ |
| | Dir(0.5) | $31.15 \pm 0.01$ | $29.41 \pm 0.23$ | $29.98 \pm 0.72$ | $28.47 \pm 0.64$ | $28.47 \pm 0.24$ | $29.17 \pm 0.28$ |
| | Dir(1) | $31.19 \pm 0.40$ | $29.77 \pm 0.25$ | $30.16 \pm 1.23$ | $29.40 \pm 0.11$ | $29.85 \pm 0.03$ | $30.20 \pm 0.32$ |
| Human-Eval | IID | $14.02 \pm 1.22$ | $7.63 \pm 2.74$ | $11.59 \pm 0.61$ | $8.85 \pm 2.13$ | $10.06 \pm 3.35$ | $9.76 \pm 1.83$ |
| | Dir(0.1) | $13.41 \pm 0.00$ | $9.15 \pm 1.83$ | $11.29 \pm 0.31$ | $10.67 \pm 2.13$ | $9.15 \pm 4.27$ | $8.24 \pm 0.31$ |
| | Dir(0.5) | $12.81 \pm 1.82$ | $9.76 \pm 1.22$ | $11.89 \pm 1.52$ | $12.19 \pm 0.61$ | $14.33 \pm 0.31$ | $8.24 \pm 1.52$ |
| | Dir(1) | $13.41 \pm 0.61$ | $7.93 \pm 1.83$ | $12.50 \pm 1.52$ | $10.37 \pm 0.00$ | $7.93 \pm 2.44$ | $7.63 \pm 1.52$ |

 **Algorithms within more DP and Non-iid Conditions**

Table 10 provides a comparative evaluation of algorithm performance when subjected to the combined effects of differential privacy and specific non-IID data distributions, namely Dirichlet distributions with $\alpha = 0.1$ representing higher data heterogeneity and $\alpha = 1.0$ indicating lower heterogeneity. This side-by-side presentation for each algorithm across various DP budgets ($\epsilon \in \{1, 3, 6\}$ and Non-Private) allows for a nuanced understanding of their robustness.

The results in Table 10 underscore FedASK's consistent ability to deliver strong performance even under these challenging compound conditions. Across the evaluated tasks (MMLU, BBH, DROP, Human-Eval), FedASK generally maintains a competitive edge or outperforms baseline methodologies for both the more heterogeneous non-IID setting $\alpha = 0.1$ and the less heterogeneous setting $\alpha = 1.0$. In particular, while increased DP noise (smaller $\epsilon$) or increased data heterogeneity (smaller $\alpha$) tends to degrade performance for all algorithms, FedASK often exhibits a more graceful degradation compared to several baselines. This suggests that FedASK's two-stage sketching and aggregation mechanism not only preserves utility under DP but also offers resilience against varying degrees of data heterogeneity. The comparative performance between the Non-IID 0.1 and Non-IID 1.0 columns for FedASK within each DP budget further illustrates its capacity to adapt effectively, reinforcing its suitability for practical federated learning scenarios where both privacy and non-IID data are prevalent concerns.

Table 10: Algorithm Performance across Varying DP Budgets for Non-IID (Dirichlet 0.1 and 1.0) Data on Llama-2-7B

| Task | DP Setting | FedASK | | FedAvg | | FFA-LoRA | | FedSA-LoRA | | FedProx | | Scaffold | |
|---|---|---|---|---|---|---|---|---|---|---|---|---|---|
| | | $\alpha$ 0.1 | $\alpha$ 1.0 | $\alpha$ 0.1 | $\alpha$ 1.0 | $\alpha$ 0.1 | $\alpha$ 1.0 | $\alpha$ 0.1 | $\alpha$ 1.0 | $\alpha$ 0.1 | $\alpha$ 1.0 | $\alpha$ 0.1 | $\alpha$ 1.0 |
| MMLU | No DP | **46.50** | **46.32** | 45.59 | 45.61 | 45.33 | 45.82 | 45.82 | 45.48 | 45.53 | 45.51 | 43.75 | 44.70 |
| | DP $\epsilon = 1$ | **45.73** | **45.88** | 42.69 | 42.12 | 41.00 | 43.75 | 41.44 | 41.62 | 41.40 | 43.66 | 43.11 | 43.60 |
| | DP $\epsilon = 3$ | **46.04** | **45.86** | 42.69 | 42.96 | 42.54 | 43.23 | 44.27 | 41.04 | 42.61 | 42.98 | 43.05 | 41.71 |
| | DP $\epsilon = 6$ | **46.24** | **46.42** | 41.79 | 43.97 | 42.82 | 42.24 | 42.94 | 43.93 | 40.07 | 43.56 | 41.06 | 43.83 |
| BBH | No DP | **32.15** | **32.55** | 33.50 | 32.03 | 32.16 | 32.86 | 32.25 | 32.11 | 33.10 | 32.29 | 32.99 | 33.08 |
| | DP $\epsilon = 1$ | **31.99** | **31.78** | 31.14 | 31.73 | 31.71 | 33.45 | 33.34 | 31.80 | 32.22 | 33.40 | 32.28 | 31.36 |
| | DP $\epsilon = 3$ | **32.46** | **32.39** | 33.54 | 31.02 | 32.71 | 32.06 | 33.37 | 32.61 | 32.30 | 32.17 | 33.02 | 31.50 |
| | DP $\epsilon = 6$ | **32.12** | **31.91** | 31.98 | 30.99 | 32.42 | 31.56 | 31.00 | 31.65 | 31.67 | 34.01 | 31.12 | 32.44 |
| DROP | No DP | **33.16** | **32.30** | 30.56 | 31.47 | 31.19 | 33.46 | 30.96 | 31.92 | 31.09 | 32.78 | 28.94 | 30.12 |
| | DP $\epsilon = 1$ | **31.93** | **31.98** | 28.18 | 29.93 | 29.75 | 31.17 | 28.96 | 28.81 | 28.16 | 30.71 | 29.45 | 29.20 |
| | DP $\epsilon = 3$ | **31.01** | **31.09** | 31.49 | 30.02 | 30.10 | 28.93 | 28.58 | 29.29 | 28.18 | 29.82 | 28.27 | 30.52 |
| | DP $\epsilon = 6$ | **32.46** | **30.90** | 28.78 | 28.35 | 29.51 | 29.21 | 30.37 | 26.51 | 29.96 | 28.51 | 28.88 | 29.80 |
| Human-Eval | No DP | **15.24** | **15.85** | 12.20 | 12.80 | 14.63 | 14.02 | 14.02 | 14.02 | 12.80 | 14.02 | 12.80 | 15.85 |
| | DP $\epsilon = 1$ | **14.02** | **12.20** | 11.59 | 10.98 | 9.76 | 12.20 | 10.98 | 10.98 | 12.20 | 12.80 | 14.02 | 12.80 |
| | DP $\epsilon = 3$ | **12.20** | **13.41** | 12.20 | 9.76 | 10.98 | 10.98 | 12.80 | 10.37 | 13.41 | 10.37 | 7.93 | 9.15 |
| | DP $\epsilon = 6$ | **12.20** | **13.41** | 6.71 | 13.41 | 10.37 | 10.37 | 6.71 | 10.98 | 11.59 | 12.80 | 14.63 | 12.80 |

## C.4 Sensitive Experiments

### C.4.1 Choice on the lora rank

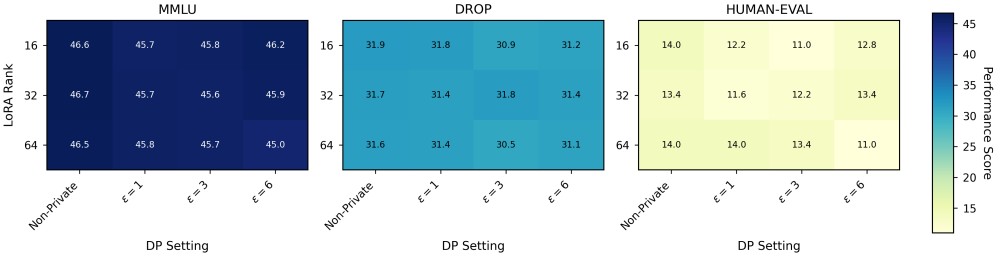

Figure 4: Performance of Llama 2-7B on IID data across LoRA ranks and differential privacy (DP) settings ($\epsilon$) for MMLU, DROP, and Human tasks.

The selection of an appropriate LoRA rank $r$ is crucial to balance the performance of the model and the efficiency of the parameters, particularly when differential privacy (DP) is applied. This

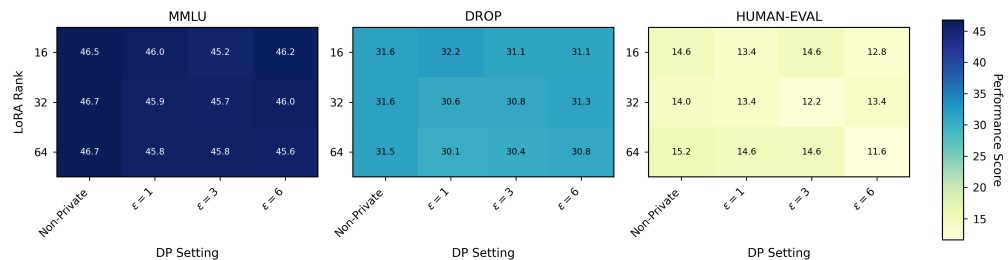

Figure 5: Performance of Llama 2-7B on Non-IID data ($\alpha = 0.1$) across LoRA ranks and differential privacy (DP) settings ($\epsilon$) for MMLU, DROP, and Human tasks.

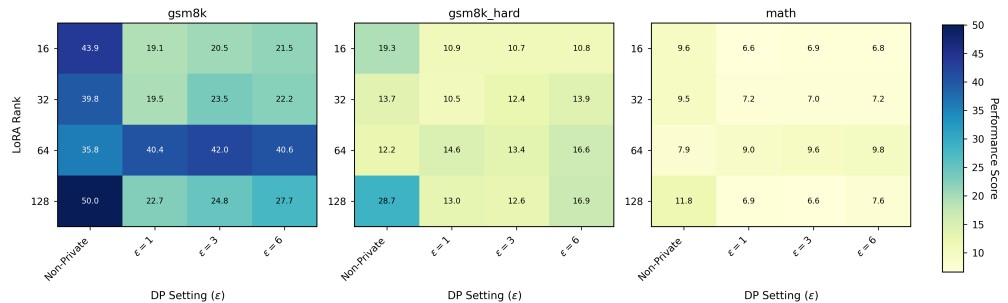

Figure 6: Performance of Llama 2-13B on IID data across LoRA ranks and differential privacy (DP) settings ($\epsilon$) for gsm8k, gsm8k_hard, and math tasks.

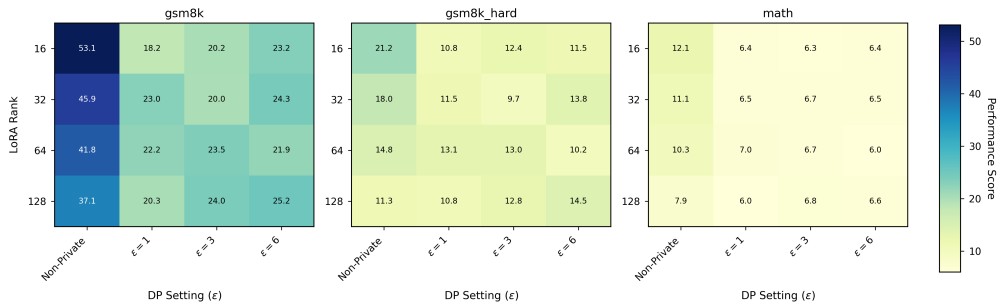

Figure 7: Performance of Llama 2-13B on Non-IID data ($\alpha = 0.1$) across LoRA ranks and differential privacy (DP) settings ($\epsilon$) for gsm8k, gsm8k_hard, and math tasks.

section details the interaction between LoRA rank, DP settings, model size, and data distribution for the FedASK framework, referencing empirical results from Llama 2-7B and Llama-2-13B models. Although higher LoRA ranks, such as $r = 128$, often deliver superior performance in nonprivate scenarios, the introduction of DP mechanisms significantly alters these performance landscapes, often favoring intermediate ranks for a more robust utility-privacy trade-off.

A key finding, illustrated in Figure 6 for the Llama 2-13B model on IID data, is the change in the optimal LoRA rank under DP for FedASK. Although rank 128 excels without privacy, for instance, on gsm8k (50.0) and gsm8k_hard (28.7), intermediate ranks frequently provide a superior utility-privacy trade-off when DP is enabled. Specifically, on the gsm8k task, rank 64 consistently outperforms rank 128 under all tested DP settings; for example, with $\epsilon = 1$, rank 64 achieves 40.4 versus 22.7 for rank 128, and with $\epsilon = 6$, rank 64 achieves 40.6 versus 27.7 for rank 128. Similarly, for the math task, rank 64 shows better performance than rank 128 across all DP budgets. On gsm8k_hard, rank 64 also remains highly competitive with, or slightly better than, rank 128 under DP conditions. This consistent strong performance of rank 64 under various DP constraints suggests that for FedASK with the 13B model on IID data, a moderately sized LoRA rank can be more parameter-efficient and achieve better utility when stringent privacy guarantees are necessary. When data heterogeneity is introduced for the Llama 2-13B model, as shown in Figure 7 for Non-IID data ($\alpha = 0.1$), the utility of intermediate ranks under DP persists largely. Although overall performance levels may adjust due to the non-IID distribution, FedASK with moderately sized ranks continues to demonstrate a strong balance between adaptation capability and resilience to DP noise, reinforcing the notion that maximal ranks are not universally optimal under privacy constraints in heterogeneous settings.

This rank-dependent performance pattern under DP is also investigated for the Llama 2-7B model. Figure 4 presents results on IID data for MMLU, DROP, and HumanEval tasks. For FedASK, it is generally observed that while larger ranks might offer marginal gains or lead in non-private scenarios, the application of DP tends to make intermediate ranks more advantageous. These moderately sized ranks appear to strike an effective balance, providing sufficient capacity for task adaptation while mitigating the detrimental impact of DP noise that can be more pronounced with a larger number of trainable parameters. The introduction of significant data heterogeneity with Non-IID data ($\alpha = 0.1$), illustrated in Figure 5, further tests this dynamic. Even in these challenging conditions, FedASK with intermediate ranks often maintains robust performance relative to larger ranks under DP. This suggests that for the 7B model, an excessively large rank under combined DP and non-IID stress may not yield proportional benefits and could be outperformed by more parameter-efficient intermediate rank configurations.

### C.4.2  Choice on over-sketching rate

The precision of FedASK's aggregation mechanism is theoretically linked to the choice of the over-sketching parameter $p$, which, together with the LoRA rank $r$, defines the sketching dimension $r + p$. While Theorem 2 provides a condition for exact aggregation, it is crucial to empirically assess the impact of varying sketching dimensions on aggregation fidelity under practical conditions, including different degrees of data heterogeneity and client participation numbers. This appendix section details these specific evaluations for FedASK, illustrating its robustness. The experiments summarized here were conducted to determine a suitable range for the sketching dimension, ensuring high fidelity without unnecessary computational overhead. All results presented pertain to the FedASK algorithm, and aggregation fidelity is quantified as the cosine similarity between the global LoRA update reconstructed by FedASK and the ideal average of true local LoRA updates.

The empirical investigation involved evaluating the aggregation fidelity of FedASK across a matrix of conditions, as depicted in Figure 9. The experiments systematically varied: (i) the sketching dimension (x-axis values: 6, 32, 51, 64, and 96), (ii) the degree of non-iid degree (Dirichlet distributions with $\alpha \in \{1.0, 0.8, 0.5, 0.1\}$), and (iii) the number of participating clients, shown in four distinct panels: (a) 5 clients, (b) 10 clients, (c) 15 clients, and (d) 20 clients. For these specific fidelity evaluations, differential privacy mechanisms were not applied to isolate the performance of the aggregation mechanism itself. The color intensity in each heatmap cell corresponds to the achieved cosine similarity, with lighter shades indicating higher fidelity.

The results consistently demonstrate FedASK's exceptional aggregation fidelity across the vast majority of tested scenarios. As seen in Figure 8, near-unity cosine similarity is achieved for most combinations of sketching dimensions, non-IID degrees, and client numbers. Even with the smallest

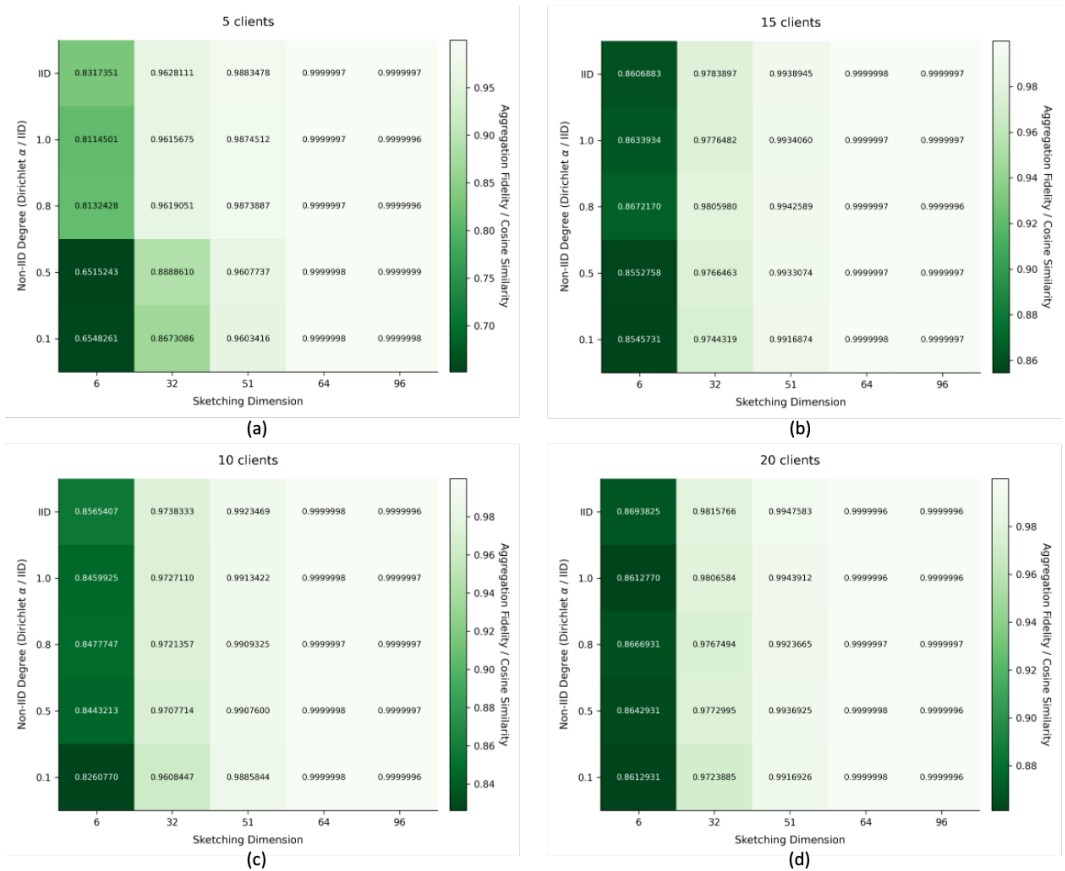

Figure 8: Impact of sketching dimension (x-axis) and non-IID degree (y-axis, Dirichlet $\alpha$ / IID) on FedASK's aggregation fidelity (cosine similarity) for (a) 5, (b) 10, (c) 15, and (d) 20 clients, showing robust near-unity performance.

sketching dimensions, fidelity remains remarkably high, particularly as the number of participating clients increases (panels b, c, and d). While the 5-client scenario (panel a) shows slightly reduced fidelity under extreme non-IID conditions and very small sketching dimensions, the performance rapidly approaches unity with modest increases in either parameter. These findings underscore that FedASK is not highly sensitive to the over-sketching rate for maintaining precise aggregation and can achieve excellent fidelity even with minimal or conservative sketching dimensions, confirming its practical efficiency and robustness.

### C.5 System Efficiency Experiments

To quantify the system efficiency of the evaluated algorithms, we conducted experiments on NVIDIA H100 GPUs, focusing on two key metrics: communication volume and end-to-end wall-clock time. The analysis uses the Llama 2-7B and Llama 2-13B models, with LoRA applied to the 'q_proj', 'v_proj', and 'k_proj' modules. Our measurements do not incorporate system-level optimizations like distributed parameter servers or communication-computation overlap.

Figure 9 illustrates the usage of resources per client. The volume of communication, segmented into uplink and downlink traffic, is reported in millions of parameters. Peak GPU memory consumption (in MB) was meticulously monitored on the client side using the `torch.cuda.max_memory_allocated(device=torch.device('cuda'))` PyTorch function, capturing the maximum memory footprint during local training operations with differential privacy mechanisms enabled. Futhermore, we analyze the communication volume per round. As shown in Table 11, the choice of numerical precision (FP16, INT8, INT4) directly impacts the data transfer size for all algorithms. The results highlight a clear trade-off: FedASK's communication volume is 25% higher than FedAvg's, but it remains significantly more efficient than other baselines,

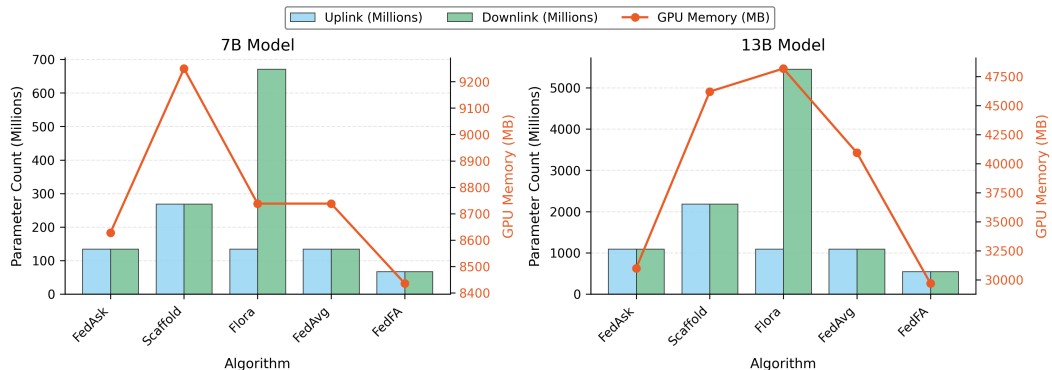

Figure 9: System resource utilization for five federated learning algorithms when training Llama 2-7B (left) and Llama 2-13B (right) models using 4-bit precision. The metrics, shown for a single client in a 5-client federated setup, include uplink and downlink communication volume (millions of parameters) and GPU memory consumption (MB).

Table 11: Total communication volume per client round for federated learning algorithms using Llama 2-7B and Llama 2-13B models under different numerical precisions. All values are in Megabytes (MB).

| Algorithm | Llama 2-7B (MB) | | | Llama 2-13B (MB) | | |
|---|---|---|---|---|---|---|
| | **FP16** | **INT8** | **INT4** | **FP16** | **INT8** | **INT4** |
| FedAsk | 1200 | 600 | 300 | 7500 | 3750 | 1875 |
| FedAvg | 960 | 480 | 240 | 6000 | 3000 | 1500 |
| Scaffold | 1920 | 960 | 480 | 12 000 | 6000 | 3000 |
| Flora | 5280 | 2640 | 1320 | 33 000 | 16 500 | 8250 |
| FedFA | 480 | 240 | 120 | 3000 | 1500 | 750 |

requiring 40% less communication than Scaffold and 77% less than the communication-heavy Flora. As demonstrated in our main paper, this modest overhead unlocks substantial performance gains, with FedASK outperforming FedAvg by up to 11.5% on MMLU and 46% on GSM8K under strong privacy.

Table 12: End-to-End Wall-Clock Time per Communication Round (s)

| Clients Num. | Alg. | Client Comp.(s) | Server Comp.(CPU) (s) | Total Comm. (s) |
|---|---|---|---|---|
| 2 | **FedASK** | 1.44 | 0.12 | 1.51 |
| | FedAvg | 1.64 | 0.07 | 1.21 |
| 10 | **FedASK** | 1.44 | 0.29 | 5.53 |
| | FedAvg | 1.64 | 0.35 | 4.84 |
| 30 | **FedASK** | 1.44 | 0.69 | 16.61 |
| | FedAvg | 1.64 | 0.71 | 13.29 |

To further assess practical performance, we empirically measured the end-to-end wall-clock time for a single FL round. The experiment fine-tunes a Llama-2-7B model, with client operations on an NVIDIA H100 GPU and server aggregation on a CPU, simulating a 1Gbps network across a varying number of clients ($K_t$).

The results, detailed in Table 12, confirm FedASK's practicality. Crucially, FedASK reduces client-side computation time by approximately 0.2 seconds per round compared to FedAvg. This is because our local DP strategy avoids backpropagation for one of the LoRA matrices. While FedASK introduces a minor computational overhead on the server, this cost is minimal and does not create a scalability bottleneck; the server time remains under one second on a CPU even with 30 clients, as the core aggregation operations do not scale with the number of clients.

