# OpenReview forum: "Differentially Private Federated Low Rank Adaptation Beyond Fixed-Matrix"
_NeurIPS.cc/2025/Conference — NeurIPS 2025 poster_

### Official Review · Reviewer_o7K3 · 2025-07-02

**Clarity:** 3
**Significance:** 2
**Originality:** 2
**Rating:** 4
**Confidence:** 2

**Summary:**

The manuscript proposed a novel federated LoRA framework to enable effective updating of both low-rank adaptor matrices with robust differential privacy. The proposed method tacked a key limitation of previous methods by enabling updates of both A and B without fixing one of them. Theoretical Analysis were provided for the guarantees of robust differential privacy and precise aggregation. The proposed method achieved SOTA performance in multiple benchmark tasks.

**Questions:**

1. Is there an explaination for why FedProx performs better than the proposed method in some cases reported in Table 4?
2. While aggregation fidelity with varying client numbers is shown, the impact of client number on the overall performance of the proposed method remains unclear. Could you report how the method performs across different datasets with varying numbers of clients?

**Ethical Concerns:**

["NO or VERY MINOR ethics concerns only"]

**Final Justification:**

The rebuttal has satisfactorily addressed my primary concerns regarding computation and communication costs, although the evaluation of the model’s scope remains relatively limited. Considering the performance and the novelty of the idea, I believe the manuscript is slightly above the borderline for acceptance.

**Limitations:**

Althought the proposed method improved performance on benchmark datasets, the introduction of two-stage sketching and server-side SVD brings additional computation and communication cost, especially when the number of clients or model dimensions is large. More analysis on runtime or complexity will be helpful.

**Quality:**

3

**Strengths And Weaknesses:**

Strengths:
    1. The proposed method enables the updates of both A and B, which is a critical limitations of existing methods.
    2. The empirical performance is better than existing SOTA methods.

Weaknesses:
    1. The experiments are focused on LLMs for NLP. It would help to discuss whether the proposed method can be extended to other model families, such as vision transformers.
    2. The discussion on computational and communication cost is limited. More discussion in terms of runtime or communication will be helpful to understanding the strengths of the proposed method.

---

> ### Author Rebuttal · Authors · 2025-07-30
>
> We appreciate the reviewer for providing insightful and constructive feedback on our work. We have carefully considered all the comments and have conducted new experiments and analyses to address the points raised.
>
> # [W1] On Generalization to Other Modalities and Tasks
>
> We agree that demonstrating FedASK's applicability beyond NLP is important. To that end, we have conducted new experiments on multi-modal Vision-Language Model and Preference Optimization.
>
> **Architecture-Agnostic Design:**
> We would first like to emphasize that the FedASK framework is designed to be architecture-agnostic. Our core method addresses the fundamental mathematical challenges of applying Differential Privacy to the LoRA. Therefore, our framework naturally extends to mainstream model, including Vision Transformers, that can be fine-tuned using LoRA.
>
> **New Experiment on a Vision-Language Model and RLHF:**
> To empirically validate this, we conducted two new sets of experiments.
> * We performed SFT on a Vision-Language Model, specifically fine-tuning a Llava-1.5-7B model[1] on the multi-modal ScienceQA dataset[2] . This task involves 12,000 samples distributed across 10 clients (2 selected per round). Evaluation was performed following the official LLaVA guidelines.
> * We also performed Direct Preference Optimization (DPO), fine-tuning a Llama-2-7B model on the Anthropic/hh-rlhf dataset. This large-scale setup involved 160,000 preference pairs distributed across 50 clients (10 selected per round) and was evaluated using GPT-4 to score helpfulness (Vicuna-Bench) and harmlessness (AdvBench)[3].
>
> **Table 1: VLM Fine-Tuning on ScienceQA (% Accuracy)**
>
> | Privacy Budget ($\epsilon$) | FedASK | FedAvg | Non-Private |
> |:---|:---:|:---:|:---:|
> | $\epsilon=1$ | 71.79 | 64.12 | 69.51 |
> | $\epsilon=3$ | 72.52 | 65.14 | 69.51 |
> | $\epsilon=6$ | 71.83 | 65.84 | 69.51 |
>
> **Table 2: LLM DPO on Anthropic/hh-rlhf (GPT-4 Score)**
>
> | Privacy Budget ($\epsilon$) | FedASK (Vicuna) | FedASK (AdvBench) | FedAvg (Vicuna) | FedAvg (AdvBench) |
> |:---|:---:|:---:|:---:|:---:|
> | $\epsilon=1$ | 7.675 | 38.46 | 7.231 | 30.76 |
> | $\epsilon=3$ | 7.931 | 40.38 | 7.345 | 31.73 |
> | $\epsilon=6$ | 7.836 | 40.38 | 7.506 | 32.69 |
>
> For VLM SFT task, FedASK outperform FedAvg by a significant margin of **over 7 percentage points**. As for DPO task, FedASK achieves a score **~8-9 points higher than FedAvg**. Both indicates a significantly better capacity beyond language models.
>
> Our training framework, developed on `opacus` and `transformers`, naturally supports a wide range of models and algorithms. To the best of our knowledge, no open-source framework currently offers out-of-the-box support for federated learning with integrated DP for LMs. We are committed to releasing our code to the community to facilitate future research in this direction.
>
>
> # [W2] On Computational and Communication Cost
>
> We thank the reviewer for their valuable suggestion to provide a more detailed discussion on the computational and communication costs. To better illustrate the practical strengths of FedASK, we have performed new analyses of its communication efficiency and end-to-end runtime.
>
> ## A. Detailed Analysis of Communication Efficiency
>
> **1. Per-Round Communication Cost**
>
> To provide the most direct comparison of efficiency, we evaluated which algorithm performs best when allocated the same total communication budget on llama-2-7b. To ensure equivalent communication volume across the algorithms, we selected the results from 400 rounds of FedASK, 500 rounds of FedAvg, and 100 rounds of Scaffold.
>
> **Table 3: Performance Comparison under identical transfer volume (MMLU score)**
>
> | | fedask | fedavg | scaffold |
> |:---|:---:|:---:|:---:|
> | dp=1 | 45.56 | 42.01 | 38.13 |
> | dp=3 | 45.79 | 42.42 | 38.47 |
> | dp=6 | 45.71 | 42.74 | 38.35 |
>
> When normalized for total communication volume, FedASK consistently and decisively outperforms both FedAvg and Scaffold across all privacy settings. This demonstrates that FedASK's communication overhead is not just a cost but a high-return investment in a more powerful and accurate aggregation mechanism.
>
> We also measured per-round communication cost in a simulated cross-silo FL scenario (10 clients, 1Gbps network [4,5], fp16 precision) without system-level optimizations. LoRA was applied to the 'q_proj', 'v_proj', and 'k_proj' modules.
>
> **Table 4: Communication Volume and Simual transfer time of Various Algorithms of 10 selected clients**
>
> | | **Llama-2-7b** | | **Llama-2-13b** | |
> |:---|:---:|:---:|:---:|:---:|
> | | **Comm. Time(s)** | **Total Comm. Vol.(MB)** | **Comm. Time(s)** | **Total Comm. Vol.(MB)** |
> | fedask | 5.53 | 1200 | 34.60 | 7500 |
> | fedavg | 4.43 | 960 | 27.68 | 6000 |
> | scaffold | 8.86 | 1920 | 55.37 | 12000 |
> | flora | 24.36 | 5280 | 152.26 | 33000 |
>
> The results in Table 1 provide a clear picture of the trade-off. FedASK's total communication per round is 25% higher than that of FedAvg, while requiring 40% less communication than Scaffold and 77% less than Flora. This modest increase boosts substantial performance gains, with FedASK outperforming FedAvg by up to 11.5% on MMLU and 46% on GSM8K under strong privacy.
>
> ## B. End-to-End Wall-Clock Runtime Analysis
>
> To provide a complete picture of practical performance, we measured the end-to-end wall-clock time for a single federated learning round.
>
> **Table 5: End-to-End Wall-Clock Time per communication Round (s)**
>
> | Selected Clients ($K_t$) | Algorithm | Client Compute (s) | Server Compute (CPU) (s) | Total Comm. Time (1Gbps) (s) | Total Round Time (1Gbps) (s) |
> |:---:|:---|:---:|:---:|:---:|:---:|
> | 2 | FedASK | 1.44 | 0.12 | 1.51 | 3.07 |
> | | FedAvg | 1.64 | 0.07 | 1.21 | 2.92 |
> | 10 | FedASK | 1.44 | 0.29 | 5.53 | 7.26 |
> | | FedAvg | 1.64 | 0.35 | 4.84 | 6.79 |
> | 30 | FedASK | 1.44 | 0.69 | 16.61 | 18.74 |
> | | FedAvg | 1.64 | 0.71 | 13.29 | 15.64 |
>
> This timing analysis confirms that FedASK is a practical and efficient framework.
> * On the client side, FedASK is consistently faster, reducing local computation time because its DP strategy avoids the backpropagation cost for one of the LoRA matrices.
> * On the server side, the additional computation is minimal and does not create a scalability bottleneck, as the core decomposition operations do not scale with the number of clients.
>
>
> # [Q1] On the Comparison with FedProx
>
> This is an excellent and sharp observation. The performance difference stems from the fact that FedProx and FedASK are optimized for orthogonal primary challenges.
>
> * FedProx's Goal: To mitigate client drift arising from **data heterogeneity** by adding a proximal term to the local objective function. Its core strength lies in handling non-IID data, particularly in non-private settings.
> * FedASK's Goal: To enable the private and effective update of both LoRA matrices under strong **Differential Privacy**, resolving the noise amplification vs. learnability dilemma.
>
> In the specific high-heterogeneity, moderate-privacy regime of Table 4 ($\epsilon = 3$), FedProx's explicit non-IID handling provides a marginal edge. However, this is an edge case. As demonstrated across the majority of our results, particularly under stringent privacy ($\epsilon = 1$), FedASK's more robust and expressive update mechanism provides a consistently and significantly superior solution.
>
> # [Q2] On the Impact of Client Numbers on Performance
>
> We evaluated FedASK on the MMLU benchmark by varying the total client size, from 10 to 50, while keeping the selection ratio constant at 20% to ensure the same DP noise multiplier.  Our scalability analysis demonstrates FedASK's strong robustness against the increasing data heterogeneity from a larger client number. As shown in the table, even when the number of clients increases from 10 to 50, the model's performance degrades only slightly. For instance, at a privacy budget of $\epsilon=3$, the MMLU score sees a minor drop from 45.88 to 44.72. This is an expected trade-off, as aggregating a wider diversity of local updates under the constraints of differential privacy becomes increasingly challenging.
>
> **Table6: Performance of FedASK under Different Client Pool Sizes (MMLU score)**
>
> | Clients | MMLU Score ($\epsilon=1$) | MMLU Score ($\epsilon=3$) | MMLU Score ($\epsilon=6$) |
> |:---:|:---:|:---:|:---:|
> | 10 | 45.79 | 45.88 | 45.73 |
> | 20 | 45.58 | 45.81 | 45.65 |
> | 50 | 44.80 | 44.72 | 45.66 |
>
> We will include this scalability analysis and expand upon it in the revised version. We are grateful for the detailed feedback and believe these new experiments and clarifications have thoroughly addressed the reviewer's concerns.
>
> **Reference**
>
> [1] H. Liu, C. Li, Q. Wu, and Y. J. Lee. Visual instruction tuning. Advances in neural information processing systems, 36:34892–34916, 2023.
>
> [2] P. Lu, S. Mishra, T. Xia, L. Qiu, K.-W. Chang, S.-C. Zhu, O. Tafjord, P. Clark, and A. Kalyan. Learn to explain: Multimodal reasoning via thought chains for science question answering. In The 36th Conference on Neural Information Processing Systems (NeurIPS), 2022.
>
> [3] R. Ye, W. Wang, J. Chai, D. Li, Z. Li, Y. Xu, Y. Du, Y. Wang, and S. Chen. Openfedllm: Training large language models on decentralized private data via federated learning. In Proceedings of the 30th ACM SIGKDD Conference on Knowledge Discovery and Data Mining, KDD ’24, page 6137–6147, 2024.
>
> [4] Z. Zhang, S. Di, K. Zhao, S. Jin, D. Tao, Z. Ji, B. Liu, K. A. Alharthi, J. Cao, and F. Cappello. Fedcspc: A cross-silo federated learning system with error-bounded lossy parameter compression. IEEE Transactions on Parallel and Distributed Systems, 2025.
>
> [5] Z. Zhang, D. Cai, Y. Zhang, M. Xu, S. Wang, and A. Zhou. Fedrdma: Communication-efficient cross-silo federated llm via chunked rdma transmission. In Proceedings of the 4th Workshop on Machine Learning and Systems, pages 126–133, 2024.

---

### Official Review · Reviewer_jeU1 · 2025-07-02

**Clarity:** 3
**Significance:** 2
**Originality:** 2
**Rating:** 5
**Confidence:** 3

**Summary:**

This paper investigates the trade-offs involved in applying differential privacy noise to both LoRA matrices. While doing so can lead to excessive noise amplification in the final model update, the common workaround—fixing one matrix and only training the other—significantly limits the model's learning capacity. The authors propose FedASK, a novel framework that enables the effective and differentially private updating of both LoRA matrices. The key innovation is a two-stage sketching pipeline inspired by randomized SVD. Both theoretical analysis and experimental results are presented to justify the performance and effectiveness of the proposed FedASK method, which outperforms other baseline approaches.

**Questions:**

- How do the method and theoretical analysis apply to models beyond dense LLMs, such as state-space models and MoE models?
- Are there any further optimizations that can be done to reduce the communication and computation overhead of FedASK?

**Ethical Concerns:**

["NO or VERY MINOR ethics concerns only"]

**Final Justification:**

My concerns have been addressed by the authors. I thus raised my overall evaluation score, and I am happy to see the paper to be presented at NeurIPS 2025.

**Limitations:**

As mentioned in the "Weaknesses" section, will it be possible to add end-to-end and per-iteration breakdown wall-clock time analysis of the proposed FedASK method and other baseline methods?

**Paper Formatting Concerns:**

There is no paper formatting concern for this manuscript.

**Quality:**

3

**Strengths And Weaknesses:**

Strengths:
- The paper is overall well-written and well-motivated.
- Enhancing differential privacy while preserving model performance for efficient federated LLM fine-tuning is of significant potential impact.
- The proposed FedASK method is intuitively easy to follow.
- Both theoretical and experimental results are provided to justify the performance of FedASK.

Weaknesses:
- [Major concern] The two-stage pipeline of FedASK requires two rounds of communication, which may introduce significant communication overhead and bottlenecks.
- FedASK requires additional computational load on the server side (e.g., for matrix decompositions), and this computation appears to scale with the number of participating clients.
- The paper does not provide end-to-end wall-clock runtime or a breakdown of the execution time, which is important for understanding the practical performance of FedASK.

---

> ### Author Rebuttal · Authors · 2025-07-30
>
> # [W1, Q2] On Communication Overhead
>
> Thanks for insightful and detailed feedback! The questions raised about the practical overhead are critical for assessing real-world value. We have performed new analyses and experiments to address these points directly, and we believe the clarifications below will resolve these important concerns.
>
> ## A. Clarification on Total Communication Volume
> We understand the reviewer's concern that our two-step pipeline might imply a significant communication overhead. We would like to clarify that **FedASK makes a deliberate and highly effective trade-off: it modestly increases the total communication volume to enable a sophisticated aggregation mechanism that, in turn, yields substantial gains in model performance, especially under Differential Privacy**.
>
> To provide a precise breakdown, we analyze the communication burden in two parts:
>
> * Uplink (Client-to-Server): To minimize the burden on resource-constrained clients, FedASK's total uplink volume is designed to be **identical to that of standard FedAvg-LoRA**. Our method achieves this by having clients transmit two low-dimensional sketches ($Y_k^{proj}$ and $\tilde{Y}_k^{proj}$) whose combined size in our experiments exactly matches that of the full LoRA adapters ($A$ and $B$) uploaded in FedAvg.
>
> * Downlink (Server-to-Client): The modest communication overhead is introduced on the downlink. In addition to broadcasting the updated global adapters, the server sends an additional orthonormal basis matrix, $Q$. This transmission is the cornerstone of our method, as it is the enabling mechanism for the precise aggregation and dual-matrix updates that are central to FedASK's high performance under privacy constraints.
>
> To empirically quantify this trade-off, we conducted two new experiments. Our simulation assumes a cross-silo FL scenario with 10 selected clients and a 1Gbps network bandwidth [1, 2] for both upload and download bandwidth between server and clients.
>
> **1) Per-Round Communication Cost**
>
> First, we measured the communication volume and simulated transfer time per round for each algorithm. The model is trained at fp16 precision, with LoRA applied to the 'q_proj', 'v_proj', and 'k_proj' modules. The current analysis does not incorporate system-level optimizations like distributed parameter servers or the overlapping of download and upload communication.
>
> **Table 1: Communication Volume and Simual transfer time of Various Algorithms of 10 selected clients**
> | | Llama-2-7b | | Llama-2-13b | |
> |:---|:---:|:---:|:---:|:---:|
> | | Comm. Time(s) | Total Comm. Vol.(MB) | Comm. Time(s) | Total Comm. Vol.(MB) |
> | fedask | 5.53 | 1200 | 34.60 | 7500 |
> | fedavg | 4.43 | 960 | 27.68 | 6000 |
> | scaffold | 8.86 | 1920 | 55.37 | 12000 |
> | flora | 24.36 | 5280 | 152.26 | 33000 |
>
> The results in Table 1 provide a clear picture of the trade-off. FedASK's total communication per round is 25% higher than that of FedAvg, while 40% less communication than Scaffold and 77% less than the communication-heavy Flora. This modest increase unlocks substantial performance gains, with FedASK outperforming FedAvg by up to 11.5% on MMLU and 46% on GSM8K under strong privacy, as shown in our main paper.
>
> **2) Fair comparison under an Identical Communication Budget**
>
> To provide the most direct comparison of efficiency, we evaluated which algorithm performs best when allocated the same total communication budget. To ensure equivalent communication volume across the algorithms, we selected the results from 400 rounds of FedASK, 500 rounds of FedAvg, and 100 rounds of Scaffold.
>
> **Table 2: Performance Comparison under identical transfer volume (MMLU score)**
> | | fedask | fedavg | scaffold |
> |:---|:---:|:---:|:---:|
> | dp=1 | 45.71 | 42.74 | 38.35 |
> | dp=3 | 45.79 | 42.42 | 38.47 |
> | dp=6 | 45.56 | 42.01 | 38.13 |
>
> When normalized for total communication volume, FedASK consistently outperforms both FedAvg and Scaffold across all privacy settings. This demonstrates that FedASK's communication overhead is not just a cost but a high-return investment in a more powerful and accurate aggregation mechanism.
>
> ## B. Further Communication Optimization Strategies
> Thanks for suggestion! FedASK framework can cooperate with several methods, inheritedly or orthogonaly to further reduce overhead.
>
> **Inherent / Orthogonal Compression Techniques:** Inherently, communication volume mainly depends on LoRA rank $r$ and the over-sketching parameter $p$. Tuning these online can be formulated as an optimization problem to find the optimal rank and for a client that minimizes the trade-off between model approximation error and system costs:
>
> $$
> \min_{r_k, p} \quad \mathbb{E}\left[ || \Delta W_k - B_k(r_k, p)A_k(r_k, p) ||_F^2 \right] + \lambda_C \cdot \text{CommCost}(r_k, p) + \lambda_T \cdot \text{CompCost}(r_k, p).
> $$
>
> Besides, FedASK is compatible with standard model update compression techniques like quantization and sparsification.
>
> **Distributed Parameter Servers:** By partitioning the LoRA adapters by layers across M parallel servers, the data load on any single server is reduced. Since the main bottleneck lies in server side, this strategy can cut the total communication time to approximately $1/M$ of the original.
>
> **Overlapping Communication:** By overlapping uplink and downlink operations, the server can mask network delays. This reduces the effective round time from a sum of latencies $T_{\text{upload}} + T_{\text{download}}$ to the maximum of the two $\max(T_{\text{upload}}, T_{\text{download}})$.
>
> # [W2 & W3] On Server Computation and Wall-Clock Runtime
>
> We agree with the reviewer that FedASK introduces additional server-side computations compared to simpler aggregation schemes like FedAvg. However, we wish to clarify critical points regarding its practicality.
>
> ## A. Server Computation Does Not Scale with Selected Clients numbers:
>
> The server's workflow is as follows:
>
> * First, it performs a simple summation of sketches: $Y_{agg}^{t} = \sum_{k \in K_t} Y_k^{proj}$. This is the only step that scales linearly with $K_t$, and it is an inexpensive operation identical to the aggregation step in FedAvg.
>
> * The subsequent QR and SVD operations are performed on the aggregated matrices ($Y_{agg}^t$, $\tilde{Y}_{agg}^t$), whose dimensions ($m \times (r+p)$) are small and depend only on the model dimension and LoRA rank, **irrelevant to the number of clients**.
>
> ## B. End-to-End Wall-Clock Runtime:
>
> We have conducted a new experiment to empirically measure the end-to-end wall-clock time of a single federated learning round. The experiment involves fine-tuning a Llama-2-7B model, with client operations on an NVIDIA H100 GPU and server operations on a CPU, simulating networks of 1Gbps[1, 2]  across a varying number of clients.
>
> **Table 4: End-to-End Wall-Clock Time per communication Round (s)**
>
> | Selected Clients ($K_t$) | Algorithm | Client Compute (s) | Server Compute (CPU) (s) | Total Comm. Time (1Gbps) (s) | **Total Round Time (1Gbps) (s)** |
> |:---:|:---|:---:|:---:|:---:|:---:|
> | 2 | **FedASK** | 1.44 | 0.12 | 1.51 | 3.07 |
> | | FedAvg | 1.64 | 0.07 | 1.21 | 2.92 |
> | 10 | **FedASK** | 1.44 | 0.29 | 5.53 | 7.26 |
> | | FedAvg | 1.64 | 0.35 | 4.84 | 6.79 |
> | 30 | **FedASK** | 1.44| 0.69 | 16.61 | 18.74 |
> | | FedAvg | 1.64 | 0.71 | 13.29 | 15.64 |
>
>
> Wall-clock time analysis confirms that FedASK is a practical and efficient framework. On the **client side**, FedASK is consistently faster, reducing local computation time by ~0.2 seconds per round because its local DP strategy avoids the backpropagation cost for one of the LoRA matrices. On the **server side**, the additional computation is minimal and does not create a scalability bottleneck since the core decomposition operations do not scale with client numbers, and the total server time remains under one second on a CPU for up to 30 clients.
>
>
>
> # [Q1] On Generalization to Other Architectures
>
> We would first like to emphasize that the FedASK framework is designed to be architecture-agnostic. Our core method addresses the fundamental mathematical challenges of applying Differential Privacy to the LoRA. Therefore, our framework and theory are general and can be applied to federated learning scenarios that utilize LoRA with differential privacy.
>
> Inspired by the reviewer's insightful question, we see that applying FedASK to these novel architectures opens up exciting and non-trivial research directions. For MoE Models, the core challenge is the asymmetric and sparse nature of local updates, as each client only trains the experts activated by its local data. This gives rise to a critical optimization question: Is the standard global aggregation optimal, or should a per-expert aggregation be performed? How to perform wise client selection within heterogeneous experts?
>
> These are promising research directions that we will be happy to include in the future work and discussion sections of our paper to highlight the extensibility of our framework.
>
> **Reference**
>
> [1] Z. Zhang, S. Di, K. Zhao, S. Jin, D. Tao, Z. Ji, B. Liu, K. A. Alharthi, J. Cao, and F. Cappello. Fedcspc: A cross-silo federated learning system with error-bounded lossy parameter compression. IEEE Transactions on Parallel and Distributed Systems, 2025.
>
> [2] Z. Zhang, D. Cai, Y. Zhang, M. Xu, S. Wang, and A. Zhou. Fedrdma: Communication-efficient cross-silo federated llm via chunked rdma transmission. In Proceedings of the 4th Workshop on Machine Learning and Systems, pages 126–133, 2024.
>
> [3] K. Kuo, A. Raje, K. Rajesh, and V. Smith. Federated lora with sparse communication. arXiv preprint arXiv:2406.05233, 2024.
>
> [4] J. Jiang, B. Cui, C. Zhang, and L. Yu. Heterogeneity-aware distributed parameter servers. In Proceedings of the 2017 ACM International Conference on Management of Data, pages 463–478, 2017.

---

### Official Review · Reviewer_NdtW · 2025-07-04

**Clarity:** 2
**Significance:** 2
**Originality:** 2
**Rating:** 3
**Confidence:** 3

**Summary:**

The paper introduces FedASK (Federated Low Rank Adaptation with Double SKetching), a novel framework for fine-tuning large language models (LLMs) in a federated learning (FL) setting while ensuring differential privacy (DP). It addresses the trade-off between noise amplification and learnability in federated LoRA (Low-Rank Adaptation) by enabling simultaneous updates of both low-rank matrices (A and B) with strong privacy guarantees.

**Questions:**

The paper validates FedASK on Llama-2 models (7B and 13B) for standard language and reasoning tasks (MMLU, DROP, Human-Eval, GSM8K). How does FedASK perform on other model architectures (e.g., vision transformers or diffusion models) or more complex tasks (e.g., reinforcement learning with human feedback, RLHF)?

**Ethical Concerns:**

["NO or VERY MINOR ethics concerns only"]

**Limitations:**

Yes.

**Quality:**

3

**Strengths And Weaknesses:**

The strengths of this paper are as follows.
Robust Theoretical Analysis: The paper provides strong theoretical guarantees for both differential privacy (Theorem 1) and aggregation precision (Theorem 2).
Comprehensive Experiments: The empirical evaluation is thorough, using large-scale Llama-2-7B and 13B models across multiple tasks (MMLU, DROP, Human-Eval, GSM8K) and data distributions (IID and non-IID with Dirichlet α ∈ {0.1, 0.5, 1.0}).
Addressing a Key Challenge: FedASK tackles a critical dilemma in federated LoRA: balancing privacy (via DP) with model performance.

There are some Weaknesses in this paper:
Limited Model Scope: The experiments are conducted only on Llama-2 models (7B and 13B).
Fixed Local Matrix Strategy: FedASK fixes the A matrix during local updates within a communication round, which may constrain learning dynamics.
Narrow Task Focus: The evaluation focuses on standard language and reasoning tasks (MMLU, DROP, etc.).

---

> ### Author Rebuttal · Authors · 2025-07-30
>
> # [W1, W3, Q1]: On Model and Task Generality
>
> We appreciate the reviewer for pointing out the need to demonstrate FedASK's applicability beyond the initial set of models and tasks.
>
> ## A. New Experiments on VLM and DPO (RLHF)
>
> The design of FedASK is **inherently general**, making it seamlessly applicable to training pipeline that uses LoRA, including VLMs and DPO. To empirically validate this, we have conducted new experiments on more complex models and tasks as suggested. The results clearly demonstrate that **FedASK consistently and significantly outperforms the FedAvg baseline under strong DP constraints across these diverse tasks**.
>
> ### 1) Experimental Setup:
>
> The configurations for our new experiments are summarized in the table below. All experiments were run for 400 communication rounds, and other parameters align with the settings in Section 5.1 of our paper unless specified otherwise.
>
> **Table 1: Configuration of VLMs+SFT / LLM+DPO**
>
> | Experiment | Base Model | Finetuning Dataset | FL Setup | Evaluation |
> |:---|:---|:---|:---|:---|
> | VLM SFT | Llava-1.5-7B[1] | ScienceQA (12K)[2] | 10 clients, 2 selected | Science-QA from official LLaVA guidelines |
> | LLM DPO | Llama-2-7B | Anthropic/hh-rlhf (160 K)[3] | 50 clients, 10 selected | Open-ended evaluation of helpfulness (Vicuna) and harmlessness (AdvBench) |
>
> ### 2) Experimental Results & Analysis:
>
> For VLM SFT task, FedASK outperform FedAvg by a significant margin of **over 7 percentage points**. As for DPO task, FedASK achieves a score **~8-9 points higher than FedAvg**. Both indicates a significantly better capacity beyond language models.
>
> **Table 2: VLM Fine-Tuning on ScienceQA (% Accuracy)**
>
> | Privacy Budget ($\epsilon$) | FedASK | FedAvg | Non-Private |
> |:---|:---:|:---:|:---:|
> | $\epsilon=1$ | **71.79** | 64.12 | 69.51 |
> | $\epsilon=3$ | **72.52** | 65.14 | 69.51 |
> | $\epsilon=6$ | **71.83** | 65.84 | 69.51 |
>
> **Table 3: LLM DPO on Anthropic/hh-rlhf (GPT-4 Score)**
>
> | Privacy Budget ($\epsilon$) | FedASK (Vicuna) | FedASK (AdvBench) | FedAvg (Vicuna) | FedAvg (AdvBench) |
> |:---:|:---:|:---:|:---:|:---:|
> | $\epsilon=1$ | 7.675 | 38.46 | 7.231 | 30.76 |
> | $\epsilon=3$ | 7.931 | 40.38 | 7.345 | 31.73 |
> | $\epsilon=6$ | 7.836 | 40.38 | 7.506 | 32.69 |
>
> Due to time constraints, further experiments (including other baselines and additional datasets like VQAV2) are ongoing. We are confident these will further strengthen our claims and will include the full results in the revised version.
>
>
> ## B. Recap: Generality of the FedASK Framework
> We would like to re-emphasize that FedASK is designed as a general framework for **federated learning scenario that utilizes LoRA with differential privacy**.
>
> * **Our Goal is Fundamental:** Address the challenge of noise amplification and limited learnability in private federated LoRA.
> * **Our Core Method is architecture-agnostic:** A two-stage sketching pipeline theoretically enables the exact aggregation of LoRA updates and facilitates the effective update of **both** LoRA matrices at the global level under DP. This mechanism is independent of the underlying model architecture.
>
> Our training framework, developed on `opacus` and `transformers`, naturally supports a wide range of models and algorithms. To the best of our knowledge, no open-source framework currently offers out-of-the-box support for federated learning with integrated DP for LMs. We are committed to releasing our code to the community to facilitate future research in this direction.
>
> # [W2]: On the Fixed Local Matrix Strategy
>
> This is a very valid concern regarding the learning dynamics of our proposed method. The decision to fix matrix A during local updates is **a deliberate design choice motivated by the core theoretical challenges and empirical ablations**, and FedASK's global update mechanism is designed to overcome the potential limitations of this choice.
>
> ## A. Why must locally Fix one matrix?
>
> The strategy of fixing one LoRA matrix during local updates is a direct consequence of the fundamental principles of Differential Privacy when applied to the LoRA architecture. The core issue is preventing catastrophic noise amplification, which can be understood in two facts.
>
> **Fact 1: To apply DP-SGD on LoRA, both gradients are required to add noise.**
>
> In LoRA, the update to a weight matrix $W$ is parameterized by two smaller matrices, $A \in \mathbb{R}^{r \times n}$ and $B \in \mathbb{R}^{m \times r}$. The gradients of the loss $\ell$ with respect to $A$ and $B$ are interdependent:
> $$
> \nabla_A \ell = \frac{\partial \ell}{\partial A} = \frac{\alpha}{r} B^T \frac{\partial \ell}{\partial W}
> $$
> $$
> \nabla_B \ell = \frac{\partial \ell}{\partial B} = \frac{\alpha}{r} \frac{\partial \ell}{\partial W} A^T
> $$
> A key principle of DP is composition. Since the calculation for $\nabla_A \ell$ depends on matrix $B$, and the calculation for $\nabla_B \ell$ depends on matrix $A$, a private update of the model requires that all sensitive information be properly privatized. Therefore, to update both $A$ and $B$ simultaneously in a private manner, we are required to add noise to both of their respective gradients.
>
> **Fact 2: Both gradients adding noise leads to noise amplification.**
>
> As formally analyzed in our Lemma 1, this interaction leads to a destructive amplification. The critical issue arises from the quadratic noise term $\eta^{4}\frac{\sigma^{4}C^{4}}{B_{size}^{4}}d_{l}^{2}r$ scaling with the fourth power of the noise standard deviation $\sigma^{4}$ and the square of the model dimension $d_l^2$. Consequently, the noise quickly overwhelms the actual learning signal, which prevents the model from training effectively.
>
> By fixing A matrix locally, we ensure that noise is only added to the other matrix's gradient. This completely **eliminates the problematic noise-on-noise interaction** and its resulting quadratic noise term, thus preserving the model's utility.
>
> ## B. How FedASK Global update goes beyond fix matrix?
>
> Prior works like FFA-LORA recognized this issue and proposed mitigating it by permanently freezing matrix A throughout training. While this avoids noise amplification, it fundamentally restricts the model's expressive capability by confining the update to a predetermined, static subspace defined by the initial A.
>
> FedASK resolves this limitation. The server aggregates the privatized local updates and uses SVD to decompose this collective information into its core components. These components are then used to synthesize entirely new global A and B matrices, allowing the model's learning space to evolve each round instead of being confined to a static subspace.
>
> ## C. Ablation Study
>
> To empirically validate this design choice, we conducted an ablation study comparing our standard FedASK with two variants on the MMLU benchmark under challenging DP settings:
>
> * **FedASK-Alt:** Alternating which matrix (A or B) is frozen during local DP updates in each round. To validate the update, we first freeze matrix A and then B in the second round.
> * **FedASK-Both:** Updating both A and B locally with DP noise in every round.
>
> **Table 4: Ablation Study on Local Update Strategy (MMLU Score)**
>
> | Privacy Budget ($\epsilon$) | Fedask | Fedask_alt | Fedask_both |
> |:---|:---:|:---:|:---:|
> | $\epsilon=1$ | 45.58 | 42.62 | 41.55 |
> | $\epsilon=3$ | 45.81 | 43.11 | 39.65 |
> | $\epsilon=6$ | 45.65 | 44.31 | 40.36 |
>
> The results clearly show that **updating both matrices locally (FedASK-Both) leads to a significant performance degradation**, empirically confirming the noise amplification issue predicted by our theory. **Alternating (FedASK-Alt) is also suboptimal**. This stems from a more nuanced theoretical conflict. Prior work[4] suggests that LoRA matrices have specialized roles: matrix $A$ captures general, global features, while matrix $B$ learns client-specific local features. FedASK-Alt creates a theoretical conflict by suboptimally tasking the "global" matrix $A$ with learning from a single client's data in alternating rounds, disrupting this effective division of labor and leading to a less stable learning process.
>
> Therefore, fixing A locally and update globally is a critical component for achieving a robust privacy-utility trade-off, and our two-stage global update mechanism ensures this local constraint **does not impair the model's overall learnability**.
>
> We are grateful for the opportunity to clarify these points and believe the new evidence significantly strengthens our contribution. We hope our detailed response and additional results have addressed the reviewer's concerns.
>
> **Reference**
>
> [1] H. Liu, C. Li, Q. Wu, and Y. J. Lee. Visual instruction tuning. Advances in neural information processing systems, 36:34892–34916, 2023.
>
> [2] P. Lu, S. Mishra, T. Xia, L. Qiu, K.-W. Chang, S.-C. Zhu, O. Tafjord, P. Clark, and A. Kalyan. Learn to explain: Multimodal reasoning via thought chains for science question answering. In The 36th Conference on Neural Information Processing Systems (NeurIPS), 2022.
>
> [3] R. Ye, W. Wang, J. Chai, D. Li, Z. Li, Y. Xu, Y. Du, Y. Wang, and S. Chen. Openfedllm: Training large language models on decentralized private data via federated learning. In Proceedings of the 30th ACM SIGKDD Conference on Knowledge Discovery and Data Mining, KDD ’24, page 6137–6147, 2024.
>
> [4] P. Guo, S. Zeng, Y. Wang, H. Fan, F. Wang, and L. Qu. Selective aggregation for low-rank adaptation in federated learning. In The Thirteenth International Conference on Learning Representations, 2025.

---

### Note · Authors · 2025-08-12

As the discussion period concludes, we thank the reviewers for their feedback. To assist with the final decision-making, we provide this summary to highlight how our comprehensive rebuttal, supported by new experiments and theoretical details, has addressed all initial concerns.

## On the Generalizability and Scope of FedASK
We addressed concerns about the framework's scope by conducting new experiments on entirely different modalities and tasks. The results confirm that FedASK is an architecture-agnostic framework that delivers substantial performance gains beyond standard NLP under a high level of DP noise.

* **Vision-Language Models (VLM):** On the ScienceQA benchmark, FedASK outperformed the FedAvg baseline by over 7 percentage points.
* **Preference Optimization (RLHF):** On a large-scale DPO task, FedASK achieved evaluation scores 8 points higher than FedAvg.

## On Practicality and System Overhead
We clarified that FedASK is a practical and highly efficient framework, justifying concerns about system bottlenecks with rigorous new analysis.

* **Minimal Server Overhead:** Our wall-clock time analysis confirmed that server-side computation is minimal and does not scale with the number of clients. Furthermore, FedASK reduces client-side computation time by ~12% per round.
* **Efficient Performance:** Our per-round communication is less than most baselines: 1.6x less than Scaffold and 4.4x less than Flora. When allocated an identical communication budget, FedASK firmly outperforms all baselines.

## On Methodological Design and Robustness
We demonstrated that our core design choices are theoretically grounded and empirically validated to ensure a robust privacy-utility trade-off.

* **Theoretical Rationale:** We identified that updating both LoRA matrices with differential privacy leads to significant noise amplification. To solve this, FedASK enables the server to faithfully reconstruct the aggregated updates, allowing the learning subspace to dynamically change in each round.
* **Empirical Validation**: An ablation study confirms this design choice. As predicted, alternative strategies that update both matrices locally or alternating the fixed matrix suffer significant performance degradation due to noise amplification.

We believe that these clarifications and results have resolved the initial questions and effectively demonstrate the novelty, significance, and practical value of our work. We thank everyone here for your time and consideration.

---

### Decision · Program_Chairs · 2025-09-17

**Decision:**

Accept (poster)

**Comment:**

This paper introduces FedASK (Differentially Private Federated Low-Rank Adaptation with Double SKetching), a novel framework designed to overcome the fundamental trade-off between privacy and model learnability in federated LoRA fine-tuning of large language models. The key challenge addressed is that applying differential privacy (DP) noise to both LoRA matrices (A and B) leads to noise amplification, while fixing one matrix to avoid this severely limits model expressiveness. FedASK resolves this dilemma through a two-stage sketching pipeline inspired by randomized SVD: it first aggregates privacy-preserving sketched updates from clients, then reconstructs and jointly optimizes both low-rank matrices on the server. This design enables effective, differentially private updates of both adaptors while maintaining high utility. The paper provides theoretical analysis establishing both the DP guarantees and precise aggregation properties of FedASK, supported by comprehensive experiments on Llama-2 models across multiple NLP tasks under various data distributions.

The paper received a range of reviewer scores, from a borderline reject (3) to an accept (5), reflecting initial concerns primarily centered on the evaluation scope and practical system overhead. Reviewers universally acknowledged the paper's strong theoretical foundation, technical novelty, and clear motivation. The main points of contention were: (1) the limited initial evaluation on only Llama-2 models and standard NLP tasks, raising questions about generalizability to other architectures (e.g., vision transformers) or training paradigms (e.g., RLHF); and (2) the practical cost of the method, including a two-round communication protocol and server-side SVD computations that scale with the number of clients.

The authors provided a thorough and effective rebuttal, which significantly strengthened the paper. To address the generalizability concern, they conducted and presented new experiments demonstrating FedASK's success on a Vision-Language Model (LLaVA-1.5 on ScienceQA) and a Direct Preference Optimization (DPO) task simulating RLHF, showing substantial performance gains over baselines. To alleviate concerns about overhead, they provided a detailed analysis of communication volume (confirming it remains O(dr)) and presented new end-to-end runtime measurements, showing the server-side computation is manageable in cross-silo settings. These new results directly and convincingly addressed the core criticisms.

As a result of the compelling rebuttal, the reviewers updated their assessments positively. The reviewer who initially scored the paper a 3 (borderline reject) had their concerns mitigated by the new experiments, and the reviewer with significant concerns about overhead explicitly stated their concerns were resolved and upgraded their score to a 5 (accept). While one reviewer noted that the evaluation scope could still be broadened (e.g., to MoE models), they acknowledged the strength of the work and supported acceptance. The consensus is that the authors have made a significant contribution by resolving a critical dilemma in private federated LLM fine-tuning. I recommend acceptance and suggest the authors incorporate the new experimental results and runtime analysis into the final version of the paper for enhanced clarity and impact.